# Recurrent evolution of high virulence in isolated populations of a DNA virus

**Tom Hill, Robert L Unckless***

The Department of Molecular Biosciences, University of Kansas, Lawrence, United States

**Abstract** Hosts and viruses are constantly evolving in response to each other: as a host attempts to suppress a virus, the virus attempts to evade and suppress the host's immune system. Here, we describe the recurrent evolution of a virulent strain of a DNA virus, which infects multiple Drosophila species. Specifically, we identified two distinct viral types that differ 100-fold in viral titer in infected individuals, with similar differences observed in multiple species. Our analysis suggests that one of the viral types recurrently evolved at least four times in the past ~30,000 years, three times in Arizona and once in another geographically distinct species. This recurrent evolution may be facilitated by an effective mutation rate which increases as each prior mutation increases viral titer and effective population size. The higher titer viral type suppresses the host-immune system and an increased virulence compared to the low viral titer type.

## Introduction

Antagonistic coevolution between hosts and their parasites is nearly ubiquitous across the diversity of life (*Burt and Trivers, 2006*). As a result, genes involved in immune defense are among the fastest evolving genes in host genomes (*Nielsen et al., 2005*; *Sackton et al., 2007*; *Enard et al., 2016*; *Shultz and Sackton, 2019*). Viruses are a particular fitness burden on hosts; for viruses to persist within populations, they must successfully invade the host organism, contend with the host-immune system, replicate and then transmit the newly produced particles to a new host (*Holmes, 2007*; *Gifford, 2012*). Once successfully established in a population, natural selection acts to modulate the rate the virus propagates relative to its virulence, optimizing the ratio of virulence to transmission (*Williams and Nesse, 1991*; *May and Nowak, 1995*; *Lipsitch et al., 1996*). Due in part to their elevated mutation rate and large population sizes, viruses can accomplish this, and often co-opt or manipulate host-pathways in the process (*Burgyán and Havelda, 2011*; *Davey et al., 2011*; *Palmer et al., 2019*).

Given the strong selective pressures viruses exert on their hosts (and hosts on viruses), it would be useful to assess how the two players behave in replicate evolutionary experiments. Experiments examining the evolution of pathogens and hosts are common in the lab (*Perron et al., 2006*; *Paterson et al., 2010*; *Bull et al., 2011*; *Martins et al., 2014*; *Scanlan et al., 2015*) and even in patients (*Pennings, 2012*; *Pennings et al., 2014*; *Feder et al., 2019*), but difficult in natural settings (*Souza et al., 2002*; *Grubaugh et al., 2015*), so we are rarely able to observe the coevolutionary dynamics between a virus and its host in replicate populations to examine natural coevolution and identify if evolution repeats itself. These studies are necessary as they frequently help characterize the pathways viruses have evolved to escape or suppress the host-immune system, and whether the same host pathways are common targets of the same virus (*Sabin et al., 2010*).

Several studies of viruses in natural populations highlight that the same initial virus can follow the same adaptive path in multiple replicate samples. This results in parallel or convergent evolution of a virus better optimized to the host, possible due to the relatively high viral mutation rate and viral population size (*Bull et al., 1997*; *Casino et al., 1999*; *Crandall et al., 1999*; *van Mierlo et al.,*

***For correspondence:**
unckless@ku.edu

**Competing interests:** The authors declare that no competing interests exist.

**eLife digest** Animals constantly evolve to protect themselves against viruses, and in turn, viruses evolve to escape their host's new defenses. As a result, genes involved in this arms' race are some of the fastest evolving in nature. A better understanding of how host-virus evolution works could help in the search for treatments for many human and animal diseases.

Repetition is one of the gold standard requirements for biological experiments. Watching different groups of animals and viruses evolve under the same conditions makes it possible for researchers to work out whether certain changes are more likely than others. This is easy to do in the laboratory, where conditions can be controlled, but much more complicated to accomplish in the wild. Wild populations are rarely completely isolated, and often face different environmental conditions. One animal-virus pair for which this is not the case is made up of the fly *Drosophila innubila*, and its virus *Drosophila innubila nudivirus*. They live in the 'sky islands' of North America, patches of forests surrounded by hundreds of kilometers of desert. These islands are like natural test tubes, isolated ecosystems each with its own separate fly and virus populations and limited gene flow between populations.

To understand how this virus-host pair evolves, Hill and Unckless sequenced the genomes of flies and viruses from four different populations. While the fly genomes did not show evidence of strong differences between populations, the virus genomes did. There were two distinct types of virus, one of which was a lot more effective than the other at infecting flies, possibly because it was better at blocking the fly's immune defenses. Unexpectedly, this virus type had evolved more than once, emerging separately on at least four different occasions. Hill and Unckless suggest that the natural interactions between flies with similar genomes and the virus guide evolution down the same path time and time again.

This work on wild populations contributes to the understanding of the evolution of viruses and their hosts. One question left unanswered is why both types of virus (one more effective at infecting the flies and the other less so) persist in each population when one is better at blocking the fly's immune response? Future work using isolated populations like these could shed more light on the pressures that shape the evolution of viruses and their hosts, potentially helping in the study of human viruses, like HIV.

*2012*). However, in other systems, the same initial virus can evolve divergently in a host or location-dependent manner. Specifically, virulence and the diversity of mutations can differ dramatically based on differences in the host environment, the extremely strong selective pressures acting on the virus, and the unpredictability of some viruses due to their elevated mutation rate (*Martinez-Picado et al., 2002*; *Cuevas et al., 2003*; *Real et al., 2005*; *Grubaugh et al., 2015*). Characterizing how viruses adapt to their long-term hosts in naturally structured populations will help broaden and expand our understanding of how hosts and pathogens evolve in response to each other, and how repeatable evolution is in the face of minor environmental differences.

We took advantage of a natural host/DNA virus infection model involving populations separated by hundreds of kilometers to study how viruses evolve with their hosts in replicate populations with limited gene flow. *Drosophila innubila* is a mycophagous species that lives in montane forests in the 'Sky islands' in southwestern North America. They are commonly infected with a double-stranded DNA virus, the *Drosophila* innubila Nudivirus (DiNV) (*Hill and Unckless, 2018*). A previous study examined the rates of evolution of DiNV (*Hill and Unckless, 2018*), finding the envelope and replication machinery to be rapidly evolving in DiNV, suggesting its importance in viral propagation (*Hill and Unckless, 2018*). Additionally, a viral suppressor of Toll was identified in the genome of a closely related virus and DiNV, suggesting that components of the Toll pathway (perhaps antimicrobial peptides – AMPs) interact with the virus, which these viruses have then evolved to suppress (*Palmer et al., 2019*). Consistent with this, genes in the Toll pathway and Toll-regulated AMPs are rapidly evolving in *D. innubila* (*Hill et al., 2019*). A more thorough examination of the natural evolution of both the host and virus is necessary to understand how DiNV and *D. innubila* interact with each other beyond this suppressor protein.

We surveyed natural genetic variation in DiNV in four replicate populations to infer its co-evolutionary history with *D. innubila*. To this end, we (a) estimated how long DiNV has infected *D. innubila*, (b) inferred the evolutionary history of the host and virus among the populations, including signatures of selection and recombination (c) examined host and virus genetic variation associated with viral titer within individual hosts, and (d) characterized gene expression differences associated with that genetic variation. We identified two viral multilocus genotypes that differ by 11 focal SNPs and found that these viral types are maintained within the same host population and across multiple isolated host populations. These SNPs are tightly linked, likely brought together by a combination of recurrent mutation and recombination, and this linkage appears to be maintained by strong selection. One viral type is associated with 100-fold higher viral titer and increased virulence compared to the other. Further, we found evidence that the SNPs associated with the high titer type have evolved independently in at least three geographically-isolated viral populations infecting the sympatric *D. innubila* and *D. azteca*, and a geographically separate population of viruses infecting *D. falleni*. Together, these results suggest rapid evolutionary dynamics of host–virus interactions, due to recurrent evolution of a highly virulent haplotype that interacts with multiple host pathways.

## Results

### DiNV segregates for linked variants strongly associated with viral titer

To characterize the evolutionary dynamics of wild *Drosophila* innubila Nudivirus (DiNV) in its host (*D. innubila*), we sequenced wild-caught individuals from four populations with the expectation that some (~40% in previous samples) individuals would be infected (*Hill and Unckless, 2020*). We considered strains to be infected with DiNV if they had at least 10x coverage for 95% of the genome. We confirmed the infection of a subsample of these using PCR for the presence of a DiNV specific gene (Supplementary Table 1). In total, we used sequencing information for 57, 92, 92, and 92 individuals from the Huachucas (HU), Santa Ritas (SR), Chiricahuas (CH), and Prescott (PR) populations with infection rates 26, 44, 71, and 79%, respectively (*Supplementary file 1* - Table 1). We also sequenced 35 individual males collected in the Chiricahuas in 2001 (52% infected with DiNV) and 80 individual males collected in the Chiricahua's in 2018 (*Supplementary file 1* - Table 2, 40 infected with DiNV and 40 uninfected determined before sequencing using PCR).

We isolated and sequenced DNA from these samples, quality filtered reads, mapped to the genome, and called genetic variation in the viral genomes to assess the extent of adaptation in each viral population. Consistent with an arms-race between host and virus, envelope and novel virulence (GrBNV-like) genes have a significantly higher proportion of substitutions fixed by adaptive evolution, compared to other viral genes (McDonald-Kreitman-based statistic Direction of Selection [*Stoletzki and Eyre-Walker, 2011*] and Selection Effect [*Eilertson et al., 2012*; *Figure 4—figure supplement 1*], DoS >0, GLM t-value >1.31, p-value<0.05).

Recurrent adaptive evolution in viral proteins known to interact with the host-immune system suggests an arms-race between *D. innubila* and DINV. To better understand the interactions between host and virus, we performed an association study between viral titer and natural genetic variation in both host and virus. We consider viral titer a proxy for virulence. For each virus-infected individual, we quantified viral titer (as the logarithm of the viral genome coverage normalized to host autosomal genome coverage) and performed an association study across both host and virus variable sites to identify variants significantly associated with viral titer using PLINK (*Purcell et al., 2007*).

Of 5283 viral SNPs in the 155kbp DiNV genome, 1,403 SNPs are segregating in at least five infected host individuals (five was our minor allele threshold for inclusion in the association study). Of those 1,403 SNPs, 78 are significantly associated with viral titer after multiple testing correction (FDR < 0.01, Significantly associated SNPs p<0.001 over 1000 permutations, *Figure 1A*). Of these, 16 are less than 2000 bp upstream of the start site of a gene, 18 are coding nonsynonymous, 11 are coding synonymous and 33 are intergenic. The SNP with the absolute lowest *p*-value for association with viral titer was also the only significantly associated SNP that seemed to segregate within individuals. In fact, the frequency of the derived allele in this nonsynonymous polymorphism in the active site of *Helicase-2* shows a negative linear relationship with the log of viral titer (*Figure 1—figure Supplement 2*, GLM t-value = −20.516, p-value=5.55e-21). However, when we ranked samples by

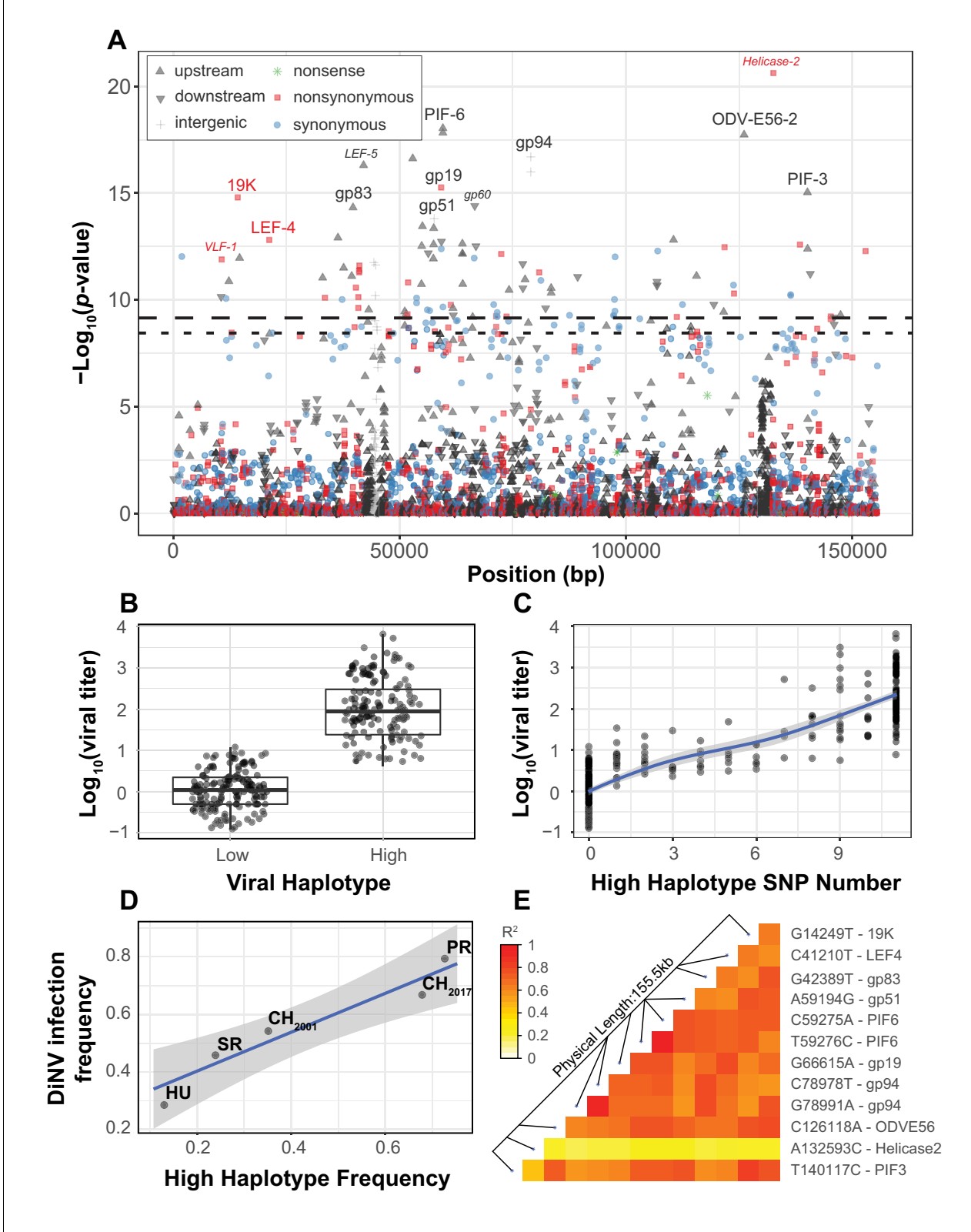

**Figure 1.** Viral genome-wide association study for DiNV titer in wild *D. innubila*. (**A**) Manhattan plot for each DiNV SNP and the significance of its association with DiNV titer. SNP point, shape and color denotes if they are upstream (black upward arrow), downstream (black downward arrow), intergenic (gray cross), synonymous (blue circle), non-synonymous (red square) or nonsense (green asterisk) mutations. Named SNPs are either part of the significantly associated viral haplotype or, if a smaller size and italicized, are in or near other genes of interest (e.g. *Helicase-2*). The FDR-corrected

*Figure 1 continued on next page*

*Figure 1 continued*

*p*-value cutoff of 0.01 is shown as a dashed line (multiple testing correction for 1403 tests), while the permutation-based genome-wide significance threshold of p=0.01 is shown as a dotted line (based on 1000 permutations). (B) Viral titer for individual wild-caught flies infected with Low and High DiNV haplotypes (containing all 11 High type alleles). The middle bar represents median value, upper and lower bars represent 25th and 75th percentile and whiskers represent a 95% confidence interval. (C) Association between the number of High type SNP variants and the viral titer of a sample. (D) Across five populations, the frequency of the High type is correlated with the frequency of the virus infection. (E) Linkage disequilibrium heatmap between the eleven focal haplotype SNPs pairwise, and with the Helicase-2 SNP. Tiles are colored by the estimated linkage between SNPs, from red (highly linked, $r^2 = 1$) to white (unlinked, $r^2 = 0$).

The online version of this article includes the following figure supplement(s) for figure 1:

**Figure supplement 1.** Frequency of each significantly associated SNP within each individual, with individuals ranked by viral titer (left = lowest, right = highest), to show the strong linkage of SNPs and little evidence of co-infection among the high and low haplotype SNPs.

**Figure supplement 2.** Permutation test for association test for viral virulence.

**Figure supplement 3.** Linkage disequilibrium between SNPs in DiNV as a heatmap generated in LDheatplot.

viral titer, we found the derived SNP frequency exhibited an almost bell-shaped curve when plotted against rank (Supplementary Figure 2).

We also identified a striking association between viral titer and eleven strongly linked polymorphisms found across the DiNV genome (*Figure 1A & D*, highlighted SNPs, *Table 1*, *Figure 1—figure supplements 1–3*, Sig. SNPs). These SNPs were significantly associated in every population when performing the association study on individual populations, or as a single group with population as a covariate. We binned strains with one of the two complete sets of alleles and referred to them as the 'High Type' and 'Low Type' (*Figure 1B*, only contains individuals infected by viruses with a complete set of 'High' or 'Low' Type SNP variants). This multilocus genotype includes three non-synonymous SNPs, five SNPs in the UTRs of known virulence factor genes and three intergenic SNPs (*Table 1*). Viral titer is, on average, 100-fold higher in individuals infected with the full High

**Table 1.** Candidate viral SNPs associated with viral titer with associated genes and the functional category of that gene.

| SNP locus | Nearest gene | SNP functional annotation | Nearest gene functional annotation | p-value (FDR-corrected) |
|---|---|---|---|---|
| G14249T | *19K/PIF-4* | Non-synonymous | Per OS Infectivity factor envelope protein, required for oral infection | 3.49e-15 (4.89e-12) |
| C41210T | *LEF-4* | Non-synonymous | RNA polymerase subunit for RNA modification | 7.99e-12 (1.12e-09) |
| G42389T | *gp83* | Upstream | Suspected virulence factor which suppresses Toll activity | 6.73–14 (9.44e-11) |
| A59194G | *gp51* | Intergenic | Suspected virulence factor | 2.96e-12 (4.15e-09) |
| C59275A | *PIF-6* | Upstream | Per OS Infectivity factor envelope protein, required for oral infection | 1.65e-18 (2.31e-15) |
| T59276C | *PIF-6* | Upstream | Per OS Infectivity factor envelope protein, required for oral infection | 1.65e-18 (2.31e-15) |
| G66615A | *gp19* | Non-synonymous | Suspected virulence factor | 3.49e-15 (4.89e-12) |
| C78978T | *gp94* | Intergenic | Suspected virulence factor | 5.93e-17 (8.32e-14) |
| G78991A | *gp94* | Intergenic | Suspected virulence factor | 1.12e-16 (1.57e-13) |
| C126118A | *ODV-E56-2* | Upstream | Occlusion-derived virus envelope protein required for particle formation | 1.43e-18 (2.01e-15) |
| A132593C | *Helicase-2* | Non-synonymous | Unwinds DNA and is critical for DNA replication | 1.11e-09 (1.56e-06) |
| T140117C | *PIF-3* | Upstream | Per OS Infectivity factor envelope protein, required for oral infection | 2.96e-15 (4.15e-12) |

type virus compared to the ancestral Low type (*Figure 1B*). When comparing the 'High' and 'Low' viral type to *Helicase-2* allele frequency, we found that viral titer increases with *Helicase-2* SNP frequency in the Low type (and some intermediate types), while titer decreases with *Helicase-2* SNP frequency for most intermediate and High types, suggesting some form of negative interaction between the High type SNP variants and the *Helicase-2* SNP (Supplementary Figure 2). Consistent with this, we found a negative interaction between *Helicase-2* SNP frequency and the presence of High type on viral titer (GLM Log10(titer) ~*Helicase-2* SNP * Haplotype: t-value = −7.815, p-value=6.39e-14).

Since these 11 SNPs are almost perfectly linked (*Figure 1E*), if one SNP were a false positive result, all would be false positives. We sought to address this in two ways. First, these SNPs are significantly associated with viral titer in every population when performing the association test separately or as a group with population as a covariate. Therefore, the likelihood of the same SNP(s) being false positives in three independent tests is quite small (p<1E-09 over 3 iterations of 1000 permutations). Second, we permuted the titer among all individuals within populations and performed the association test (100,000 total permutations). For each permutation, we took the SNP with the lowest *p*-value from each permutation test and the 10 other SNPs most strongly linked to it. We then compared the viral titer in those with this permuted 'High' haplotype to those with the 'Low' haplotype. In no case were the permuted differences between High and Low haplotypes as large as those observed from the real data (*Figure 1—figure supplement 2A*, 100,000 permutations), and the distribution of true *p*-values diverges dramatically from the permuted null expectation (*Figure 1—figure supplement 2B*, black). We found this divergence from the null expectation of *p*-values disappears when including viral haplotype as a covariate (*Figure 1—figure supplement 2B*, red), suggesting that the High type drives most of the signal seen in the association study.

Though we found few strains with an intermediate number of SNPs (intermediate types), viral titer appears to increase as the number of High type SNPs increases (*Figure 1C*, GLM t-value = 34.971, p-value=5.912e-16), but the rate of increase slows as the number of High type SNPs increases suggesting diminishing returns (*Figure 1C*). Some of these polymorphisms are associated with known virulence factors, or are related to the formation of the viral envelope co-opting the host vesicle trafficking system and are rapidly evolving in nudiviruses (e.g. *19K, ODV-E56, PIF-3*) (*Rohrmann, 2013*; *Hill and Unckless, 2017*; *Hill and Unckless, 2018*). Additionally, several are associated with genes exclusive to a few nudivirus genomes thought to be novel virulence factors, including *gp83,* a gene that downregulates Toll-induced antimicrobial peptides (AMPs) and upregulates those induced by IMD (*Palmer et al., 2019*). Both the Toll and IMD pathway may interact with DNA viruses (*Zambon et al., 2005*; *Costa et al., 2009*; *Merkling and van Rij, 2013*; *Ferreira et al., 2014*; *Lamiable et al., 2016*; *Palmer et al., 2019*).

Among populations, we found a positive correlation between the frequency of the High type and overall DiNV infection frequency (*Figure 1D*, GLM logistic regression z-value = 6.104, p-value=0.00883), suggesting that the High type may have a higher transmission rate than the Low type, resulting in a larger number of new individuals infected per DiNV-infected individual. We also found both viral types in collections from 2001 and 2017, with the High type significantly more common in the 2017 collection (Fisher Exact Test p-value=0.0167, *Figure 1D*).

It is possible that mutations could segregate within individuals, but this is relatively rare overall, and we found no evidence of the 11 haplotype SNPs segregating within any individual fly (Supplementary Figure 2). In fact, no individual fly contains more than two significantly associated SNPs segregating within the sample, suggesting that hosts are either infected completely with Low type or High type virus particles.

## The high DiNV viral type more effectively suppresses the Drosophila immune system

Given the striking difference in viral titer between High and Low type viruses, we sought to further characterize the differences in infection dynamics between the types, focusing on the differences in expression between viral types. We sequenced mRNA from 80 wild *D. innubila* males collected in 2018 (*Supplementary file 1* - Table 2, 40 infected with DiNV, 40 uninfected) and performed a differential expression analysis between infected and uninfected individuals. Few genes were differentially expressed (DE) between infection states in *D. innubila*, but these DE genes were enriched for several interesting categories. Specifically, we found IMD-induced antimicrobial peptides (AMPs) were

upregulated upon DiNV infection, while one Toll-induced AMP, and several chorion and heat shock protein genes were downregulated (*Figure 2—figure supplement 1A*). We also compared these results to a laboratory experiment in *D. melanogaster* of differential expression after infection with a close relative of DiNV (*Palmer et al., 2018*). These same genes tend to also be differentially expressed in *D. melanogaster*. Of the 12 genes which are differentially expressed in the same direction in both species, five are AMPs and five are chorion genes (*Figure 2—figure supplement 1B*). This suggests that even with significant evolutionary divergence in both host and virus, the transcriptional response to infection is similar in both hosts. It was unexpected that chorion proteins are differentially expressed upon infection by DiNV and Kallithea virus (*Figure 2*, *Figure 2—figure supplement 1*), especially as DiNV is thought to infect the Drosophila gut (*Unckless, 2011*) and these samples are male. This suggests DiNV may infect more tissues than just the gut, and that chorion proteins are expressed in the germline of both *D. innubila* sexes, potentially fulfilling different roles in different sexes/species.

To determine how the High and Low type differ in their ability to infect *D. innubila,* we compared gene expression between *D. innubila* infected with the two types from this same collection (excluding intermediate types, which also showed a significant difference in viral titer between High type and Low type, p-value=0.0000143). We found 17 host genes and nine viral genes differentially expressed between types (after we controlled for virus copy number as FPKM/titer for viral genes *Figure 2A*, FDR-corrected p-value<0.01). Specifically, three Toll-regulated immune peptides (*IM33*, Bomanins *BomBC2,* and *BomT2*) and one JAK-STAT regulated immune peptide (*Listericin*) have reduced expression in High type infected individuals compared to the Low type (*Figure 2*, *Figure 2—figure supplement 2*). Viral genes of interest (*PIF-3, 19K, gp83*) have higher expression per viral particle (FPKM/titer) in the High type compared to the Low type. *gp83* also increases in expression per viral particle (FPKM/titer) as the number of High type alleles increases (*Figure 2B*, t-value = 13.732, p-value=3.36e-15). Together these results suggest that the High type has increased expression of key virulence factors, which in turn, manipulate the expression of host genes involved in immune defense to result in the observed differences in viral titer. Specifically, higher *gp83* expression may cause the lower Toll-mediated AMP expression (*Palmer et al., 2019*). We reasoned that if the High type is suppressing the host Toll pathway, *Myd88*, the Toll signaling protein upstream of AMPs would also have lower expression. We did find *Myd88* expression is lower in strains infected with the High type (*Figure 2A*, though no significantly so), which in turn might prevent the host from enacting a proper immune response to DiNV infection (*Figure 2*, *Figure 2—figure supplement 1*).

## Experimental infections recapitulate differences in viral type virulence

To assess if the virulence differs between virus types, we performed experimental infections of *D. innubila* males using viral filtrate of strains infected with one of the two types of DiNV. Before injections, we used qPCR to identify the viral titer of each sample and dilute all samples to the same viral concentration. In these experiments, as viral titer increases, the survival of infected flies decreases, regardless of viral type (*Figure 3—figure supplements 1* and *2*, ANOVA residual deviance = 3.536, p-value=2.454e-07, Cox Hazard Ratio z-value >2.227, p-value<0.02592). In both types viral titer also increased for the first 3 days of infection (GLM t-value = 9.817, p-value=3.6e-14). This established that viral titer is a reasonable proxy for virulence and that DiNV is virulent in *D. innubila*.

For a set of four High type and four Low type-infected individuals, we isolated viruses, diluted to roughly equal concentrations of viral particles and performed infections for replicates of 10 males with microneedles dipped in one of the filtrate samples. Survival is significantly lower for flies infected with High type viruses when compared to either flies pricked with sterile media (*Figure 3A*, Cox Hazard Ratio z-value = 3.671, p-value=0.000242) or those pricked with Low type virus (*Figure 3A*, Cox Hazard Ratio z-value = 4.611, p-value=4.02e-06). Flies pricked with Low type virus show a non-significant reduction in survival compared to control flies (Cox Hazard Ratio z-value = 1.353, p-value=0.176). We also measured viral titer over time using qPCR, and found titer increases through time in flies infected with either type (*Figure 3B*, GLM $Log_{10}$(titer) ~days + type + vial | strain, days t-value = 9.912, p-value=1.76e-14). Flies infected with High type virus have significantly higher viral titer compared to flies infected with Low type virus (*Figure 3B*, GLM $Log_{10}$(titer) ~days + type + vial | strain, type t-value = 3.934, p-value=0.000211). These results suggest that the higher viral titer observed in the High Type is also associated with higher virulence.

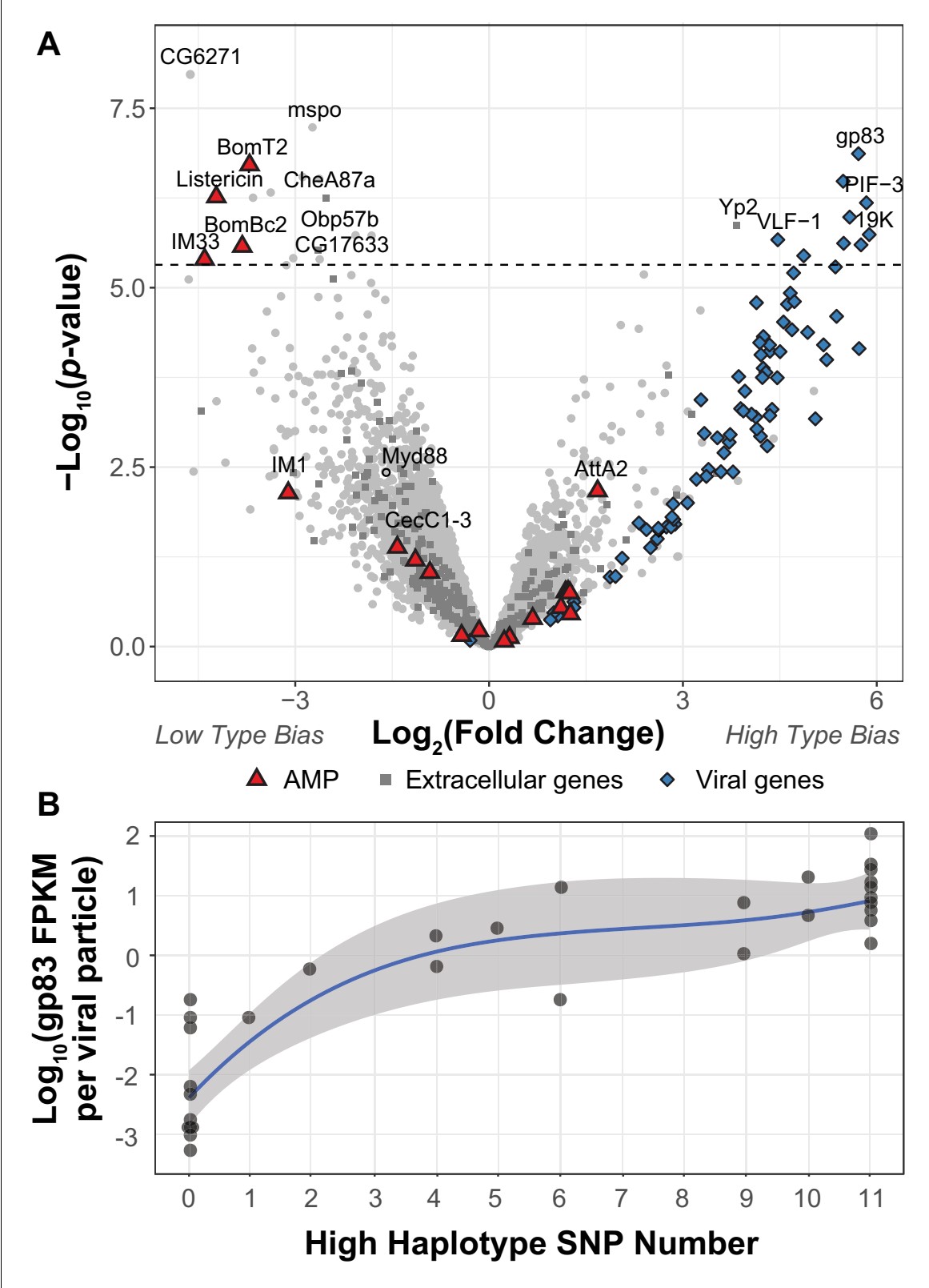

**Figure 2.** DIfferential expression between two viral haplotypes. (**A**) Differential expression of *D. innubila* and DiNV genes between *D. innubila* infected with either the Low type or High type DiNV multilocus genotypes. For host genes, the log-fold change of mRNA fragments per million fragments is compared, while for viral genes the log-fold change of viral mRNA fragments per million fragments per viral particle is compared. Genes are colored/labelled by categories of interest, specifically antimicrobial peptides (AMPs), proteins involved in the extracellular matrix and viral proteins. Specific

*Figure 2 continued on next page*

*Figure 2 continued*

genes of interest, such as *Myd88*, are also named. The FDR-corrected significance cut-off of 0.01 (10,320 tests) is shown as a dashed line. (B) Expression (in FPKM per viral particle) of *gp83* increases with the number of High type SNPs.

The online version of this article includes the following figure supplement(s) for figure 2:

**Figure supplement 1.** Volcano plot of changes in gene expression between *D. innubila* infected with DiNV and uninfected controls.

**Figure supplement 2.** Expression changes (shown as transcript fragments per 1 million reads per 1kbp of exon) of antimicrobial peptides between strains infected with High type DiNV, Low type DiNV or not infected.

## DiNV types are under strong selection in the host

We next tested whether genes likely to be involved in host/virus interaction show signs of recurrent natural selection. Using McDonald-Kreitman based statistics for estimating the proportion of substitutions fixed by selection (*McDonald and Kreitman, 1991*; *Stoletzki and Eyre-Walker, 2011*; *Eilertson et al., 2012*), we tested whether genes that are associated with the High and Low types exhibited different signatures of natural selection compared to other viral genes. We calculated the Selection Effect, the proportion of substitutions fixed by adaptive evolution, weighted by the total number of substitutions in the genome (*Eilertson et al., 2012*). We found that genes associated with SNPs in the initial association study for viral titer, which defined the High and Low types (listed in *Table 1* and including *19K, PIF-3* and *LEF-4*) have significantly higher rate of substitutions being fixed due to selection than background genes across all categories (*Figure 4*, type-associated genes versus all other, t-value = 2.718, p-value=0.00068). When separating the polymorphisms from the substitutions, we found that envelope genes have a significant excess of functional substitutions per site compared to other genes (*Figure 4—figure supplement 2A*, GLM t-value = 3.62, p-value=0.00107), while genes of unknown function have a significant excess of non-synonymous polymorphisms (*Figure 4—figure supplement 2A*, GLM t-value = 2.33, p-value=0.02241) and a

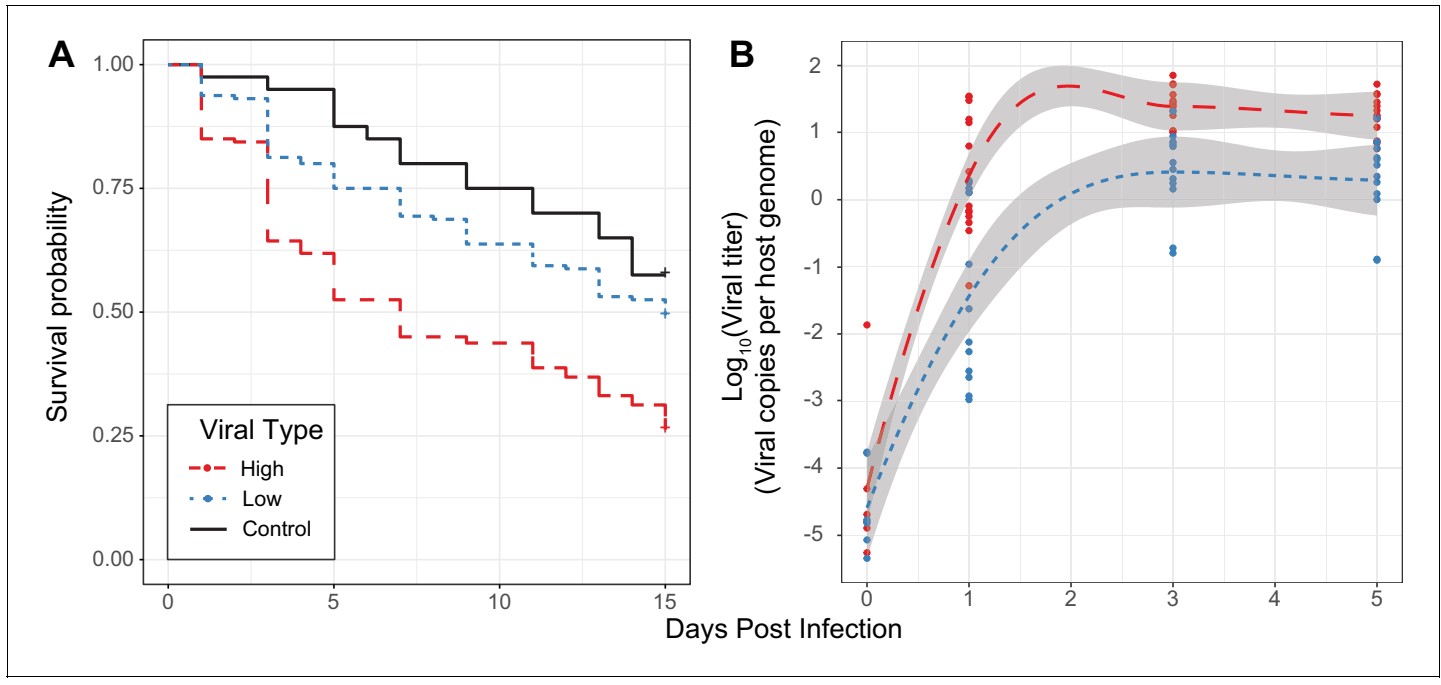

**Figure 3.** Effect of viral type in experimental infections. (A) Survival curves of *D. innubila* infected with high and low viral types compared to control flies pricked with sterile media, for 15 days post-infection. Survival 5 days post-infection separated by strain is shown in *Figure 3—figure supplement 2*. (B) qPCR copy number of viral *p47* relative to *tpi* in *D. innubila* infected with DiNV filtrate of high and low types.

The online version of this article includes the following figure supplement(s) for figure 3:

**Figure supplement 1.** Effect of differences in viral type and titer in experimental infections.

**Figure supplement 2.** Survival of flies following infection with different titers and types of DiNV.

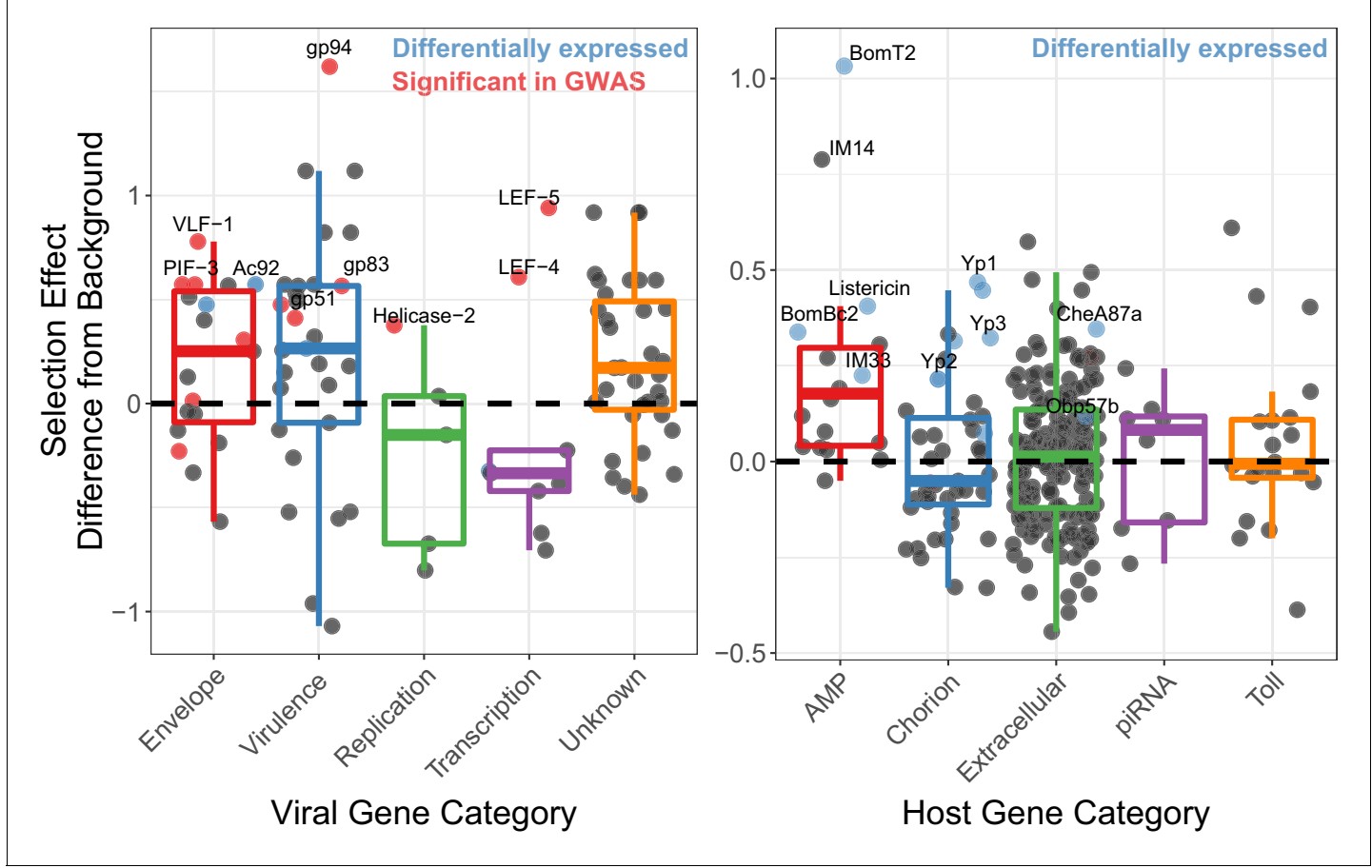

**Figure 4.** Genes implicated in host/virus interaction are rapidly evolving by positive selection in the Chiricahua population. Difference in selection effect for viral and host gene categories of interest from nearby background genes (average shown as 0, the dashed line), as indicated by the proportion of substitutions fixed by adaptation, weighted by mutations in SnIPRE (*Eilertson et al., 2012*). Genes that have associated SNPs from the association study are highlighted in red, while genes which are differentially expressed upon infection, or between viral types are labelled in blue. All association study hits are also differentially expressed and labelled in red. Genes of interest are named.

The online version of this article includes the following figure supplement(s) for figure 4:

**Figure supplement 1.** McDonald-Kreitman based statistics for each gene in population of *Drosophila* innubila Nudivirus, with viral envelope and GrBNV potential virulence factors shown separately.

**Figure supplement 2.** The ratio of nonsynonymous to synonymous polymorphisms and substitutions for both DiNV and *D. innubila*, used to generate *Figure 4*.

**Figure supplement 3.** Host genome-wide association study for DiNV titer in wild *D.innubila*.

deficit of non-synonymous substitutions (*Figure 4—figure supplement 2A*, GLM t-value = −3.894, p-value=9.93e-05), which will lower the average selection effect of background genes, suggesting that the relative excess adaptation seen in some genes is in part due to an excess of variation in genes on unknown function.

We also performed an association study using the host polymorphism and found 13 significantly associated SNPs, after controlling for the viral haplotype (*Figure 4—figure supplement 3*, p<0.01), but found no significant enrichments (p-value>0.05) or genes of interest (e.g. those involved in Toll signaling or antiviral pathways). We therefore looked for enrichments in genes associated with the top 100 significantly associated SNPs versus all other genes and found piRNA genes enriched (GO enrichment = 3.44, p-value=0.035, FDR-corrected). Given that siRNA genes are not highly ubiquitously expressed in *D. innubila* (*Hill et al., 2019*), that DiNV reduces host fecundity (*Unckless, 2011*), and that a close relative of DiNV infects the host ovaries (*Palmer et al., 2018*), DiNV could interact with chorion proteins during oogenesis, and piRNAs could be suppressing DiNV (*Lewis et al., 2018*). Consistent with the arms race model, host genes we suspect are interacting with DiNV (such

as the association study hits, AMPs, chorion genes, piRNA genes, and extracellular genes) show elevated levels of substitutions fixed by selection compared to background genes in *D. innubila* (*Figure 4* and *Figure 4—figure supplement 1*, GLM p-value<0.05) (*Hill and Unckless, 2020*). Finally, differentially expressed chorion genes, extracellular genes and AMPs have significantly more adaptive substitutions than non-differentially expressed genes in the same categories (*Figure 4*, blue dots, differentially expressed versus all other T-test: *D. innubila* t-value = 4.755, p-value=0.000671). When separating the polymorphisms from the substitutions, we found an excess of functional substitutions per site in AMPs (*Figure 4—figure supplement 2B*, GLM t-value = 4.776, p-value=1.81e-06), driving their excess of adaptive substitutions compared to other genes. Overall, these results suggest strong selection is acting on both the host to suppress viral activity and the virus to escape this suppression.

## Recombination in DiNV may facilitate the evolution of the high type

We were interested in examining the genetic variation of DiNV to determine the relationship of the host and the two putative types of DiNV. To determine the appropriate approaches for measuring these patterns, we first need to determine the effective rate of recombination in the virus. Though recombination is necessary for proper nudivirus replication (*Kelly, 1982*; *Kamita et al., 2003*; *Rohrmann, 2013*), often these recombination events will be between nearly identical viral particles resulting in no detectable signature of recombination across the genome (*Rohrmann, 2013*). For recombination to leave its signature in genetic variation, two divergent viruses must coinfect the same cell – we refer to this as effective recombination. It is unclear how common such recombination is in nudiviruses.

We used three methods to examine rates of effective recombination across the DiNV genome: First we used GARD to identify the number of recombination breakpoints across samples (*Kosakovsky Pond et al., 2006*). Second, we screened for recombination events by finding all four combinations of alleles between two SNPs. Third we calculated the linkage disequilibrium pairwise between all SNPs (*Shin et al., 2006*). We found recombination is relatively common in DiNV (*Figure 1—figure supplement 3*), with 307 potential recombination events genome-wide in recent history (based on combinations of alleles across our samples), and that the genome has relatively low linkage disequilibrium (*Figure 1—figure supplement 3*, *Figure 5—figure supplements 1* and *2*). In DiNV, the eleven SNPs significantly associated with viral titer (the High and Low types), are spread across the ~155 kilobase pair genome, yet are nearly perfectly linked to each other but not to other SNPs (*Figure 1—figure supplement 3*, *Figure 5—figure supplements 1* and *2*).

We wanted to examine if it was possible that the High type was generated by recombination of SNPs onto the same background, under the assumption that if the High type was formed via recombination we would expect to find recombination events each side of each significantly associated SNP. Using GARD and the four-allele test, six of the significantly associated SNPs show evidence of a recombination event on one side of the SNP, while three have evidence of a recombination event on each side of the significantly associated SNP (*Figure 5—figure supplement 3*, Supplementary Data). This suggests that recombination could have aided in the formation of the multilocus genotype, by recombination between two intermediate types to form the High type. However, we found no evidence of recombination between individuals in different populations (e.g. no recombinant haplotypes that are ½ CH and ½ PR), suggesting little movement of one or more significantly associated SNPs between populations. We also found no evidence of recombination between the complete High and complete Low types. This suggests that the initial SNPs for the High type have evolved on separate backgrounds in all three populations, though recombination between intermediate strains may have allowed for the formation of the complete High type, or for the generation of other intermediate types. For this to happen, each SNP may have recurrently evolved in each population, even if not sequentially.

## The high viral type of DiNV evolved repeatedly in three *D. innubila* populations

We next sought to understand the evolutionary origin of the two putative types. Given that both types are found in all populations surveyed (*Figure 1D*) with no evidence of recombination between populations, we hypothesized that this could occur one of three ways: First, the derived haplotype

was present ancestrally and has been maintained since before geographic isolation occurred. Second, the derived haplotype evolved following geographic isolation and has spread via migration between locations. Third, the derived haplotype has recurrently evolved in each location.

To distinguish between these possibilities and determine the timeframe of divergence, we used the site frequency spectrum of silent DiNV polymorphism to estimate effective population size backwards in time for all populations (*Liu and Fu, 2015*). We found that the three populations (CH, HU and SR) expanded from a single viral particle ($N_e$ = 1) to millions of particles during the last glacial maximum (30–100 thousand years ago) when *D. innubila* settled its current range (*Figure 5—figure supplement 4*; *Hill and Unckless, 2020*). This supports a single invasion event during a host-range change for each location. PR appears to expand between 1 and 10 thousand years ago, suggesting a much more recent bottleneck during the range expansion in PR (*Figure 5—figure supplement 4*; *Hill and Unckless, 2020*).

We aligned genomic regions containing SNPs to two related nudiviruses, Kallithea virus and Oryctes rhinoceros Nudivirus (OrNV) (*Wang et al., 2008*; *Hill and Unckless, 2018*; *Palmer et al., 2018*). The High type alleles are not present in either Kallithea or OrNV, and are not found in short read information for wild *D. melanogaster* infected with Kallithea virus (*Webster et al., 2015*), suggesting they are derived in DiNV.

We generated consensus DiNV sequences for each infected *D. innubila* individual and created a whole-genome phylogeny to infer geographic diffusion of samples using BEAST2 (*Bouckaert et al., 2014*). We then performed ancestral reconstruction of the presence of the High type across the phylogeny using APE (*Paradis et al., 2004*). Our samples grouped as three populations (with HU and SR forming one population) and, consistent with our expectation, the Low type is the ancestral state (*Figure 5A*). Interestingly we found PR is nested within HU/SR, based on five SNPs which segregate in the Low type HU/SR but are fixed in all PR samples (*Figure 5A*), consistent with the expansion of DiNV north over time. Surprisingly, the High type appears to have evolved repeatedly and convergently within each population, forming separate groups within each population (*Figure 5A*). The High type also clusters within each population in a phylogeny or principal component analysis of all viral SNPs (*Figure 5—figure supplement 1C*), and when repeating these analyses while excluding the eleven focal SNPs.

We next surveyed each background SNP (e.g. SNPs not associated with the High or Low type) to determine if the general background supports one of the three outlined ways in which the High type evolved and spread in each location. We grouped SNPs by their presence in just the High type or Low type (supporting a single origin and spread by migration) or if they were unique to a single population but shared between the High and Low types. In total, 341 SNPs (24% of SNPs surveyed) are unique to a single population yet are still shared between both High and Low types (*Figure 5* and *Figure 5—figure supplement 1*), compared to 23 SNPs (including the 11 High type SNPs) shared between locations but exclusive to High type samples. This is consistent with the lack of recombination between individuals in different populations, suggesting the High type associated SNPs are derived recurrently in each population on different backgrounds. Consistent with this, using Tree-Time we found that the eleven significantly associated SNPs could have recurrently evolved across our samples, after accounting for recombination and mutation rate (*Kosakovsky Pond et al., 2006*; *Sagulenko et al., 2018*).

We identified 341 SNPs (161 for CH, 127 for HU and SR, and 53 for PR), are present in all High type samples of a single population but a variable proportion of Low types for that population (between 19 and 94%) and are unique to that population. This pattern fits with the High type recurrently evolving on a single background (a different background in each location), supporting recurrent evolution of the High type, and against two ancestrally maintained viral types. The population-specific background SNPs are spread throughout the DiNV genome, with little evidence of recombination with the High type SNPs, making it unlikely that these SNPs recombined onto different backgrounds, and were instead present when the High type first evolved in each population (*Figure 5—figure supplements 1* and *2*). We found little evidence of gene conversion producing this background signature (*Figure 5—figure supplement 3*). While we identified recombination events between background SNPs in the Low type, we found no evidence of this occurring due to their fixed states in the High type, likely because the High type has swept to higher frequencies within individuals before recombination can occur.

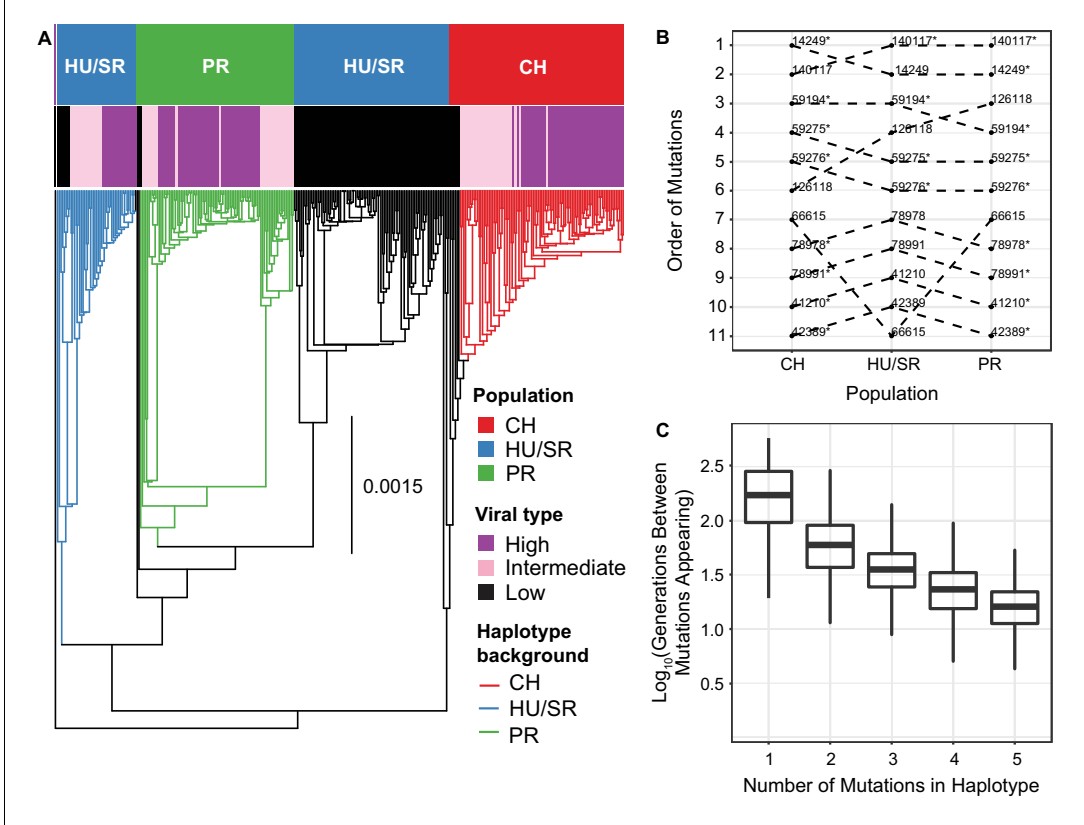

**Figure 5.** The evolution and maintenance of two viral types. (**A**) Phylogeographic reconstruction of the spread of DiNV through *D. innubila,* rooted on the Kallithea virus reference sequence, including a reconstruction of the High type evolution (with strains containing all 11 High type variants shown in purple, strains with an intermediate number of high type variants are shown in pink, and strains with no high type variants are shown in black). Branches are colored when the SNPs found in the background for each High haplotype are present in the population, showing that the background differs per population. Black branches show states where branch tips do not contain all the shared High/Low population specific background SNPs. (**B**) Order of mutations in the viral haplotype appearing in each population. Apart from three mutations the order is consistent between locations. SNP order has been bootstrapped across multiple sample phylogenies, SNPs with bootstrap support of >95% are labelled with a *. (**C**) The number of generations needed for 'High titer' mutations to evolve in simulated populations, given that each mutation increases the mutation rate. The number of generations between each mutation appearing decreases as titer increases.

The online version of this article includes the following figure supplement(s) for figure 5:

**Figure supplement 1.** SNP association in the viral haplotype and principle component analysis of viral samples.

**Figure supplement 2.** Linkage ($r^2$) between different types of SNPs in each population of DiNV, and across all samples.

**Figure supplement 3.** Associations of significantly associated SNPs and their neighboring SNPs, to attempt to determine if gene conversion is the cause of the recurrent evolution of the viral haplotype.

**Figure supplement 4.** Effective population size backwards for each population of DiNV going backwards in time, estimated using StairwayPlot.

## The elevated DiNV mutation rate associated with increased titer allows the high type to recurrently evolve in each population

Though there is strong linkage between the High type SNPs, they are not perfectly associated with each other (*Figure 5—figure supplements 1* and *2*). Using this slight disassociation and APE (*Paradis et al., 2004*), we performed ancestral reconstruction of SNP origins in each population (assuming recurrent evolution) and found that, excluding three variable SNPs, the evolution of these SNPs was a similar order in each population (*Figure 5B*).

The recurrent evolution of these mutations in a specific order is feasible as the mutation rate is high and the waiting time between mutations should decrease as titer increases. Logically, if the wait time for the appearance of a beneficial mutation is $1/(2N_e*\mu*s)$, increasing $N_e$ (with titer) should decrease the wait time for the appearance of the High type mutations (*Gillespie, 2004*). To determine if this recurrent evolution is plausible in our estimated timeframe (~10,000 years), we simulated

viral populations using a discrete susceptible-infectious model implemented in deSolve (*Soetaert et al., 2010*) using estimated baculovirus mutation rates, recombination rates, ranges of viral titer taken our samples and estimated population sizes for each viral population (parameters described in the methods). In this model we used an effective mutation rate scaled to viral titer, considering the mutation rate per particle, so total mutations per generation increase with viral titer. For simplicity, we limited these simulations to the first five mutations as these are mutations of largest effect and appear to be necessary for the remaining six mutations to appear. We also included epistasis which reduces the increase in titer for each subsequent mutation ($titer^{\sqrt{no.\ muts}}$), as a similar reduction is seen in our samples (*Figure 1C*). The simulations suggest that waiting time for the first mutation that increases titer is highly variable between replicates but usually occurs within 1000 generations (~200 years at most, assuming the virus is transmitted from adult hosts to larval hosts via feces, with a 5 host generations per year as a conservative minimum estimate of viral generations per year, in >99.93% of replicates, *Figure 5C*). The average wait time for each subsequent mutation decreases monotonically (GLM t-value = -2.389, *p*-value = 0.03686). In most cases, the next mutation appears in the background of the previous high titer mutation (*Figure 5C*) due to the elevated effective mutation rate and increased basic reproduction number ($R_0$). In 34 of 1000 simulations, when a mutation does appear on a different background, recombination facilitates the generation of the full complement of mutations. Given that effective mutation rate is much higher than the effective recombination rate following the appearance of the first mutation (as recombination does not scale with titer in nudiviruses and baculoviruses *Kamita et al., 2003*), this likely accounts for the mutation rates dominance in our simulations, though recombination likely does play a role in recombination between intermediate strains. The accumulation of mutations occurs at close to a geometric (approximately exponential) rate. Additionally, the standard deviation of time wait times also decreases with each new mutation (GLM t-value = -2.441, *p*-value = 0.04241), increasing the certainty that the entire multilocus genotype will appear in a population rapidly once the initial mutations appear. This chain reaction of adaptation could easily facilitate the repeated evolution of the virulent High type independently in three populations, with all eleven mutations fixing in a population within 6000 generations (~1200 years maximum) in all replicates (3372 generations on average, ~675 years maximum), a plausible amount of time given our estimated timeframe.

## Both viral types are found in two other *Drosophila* species and have also evolved in a geographically distinct population

Since we found two types of DiNV are present in all *D. innubila* populations, and that other species are infected with DiNV (*Unckless, 2011*), we hypothesized that another species could be a reservoir for the less virulent Low type. We chose to study *D. azteca* from the Chiricahuas since it is frequently infected with DiNV (~33% infection), overlaps with *D. innubila,* and is genetically divergent (40–60 million years) which could mean a very different genetic interaction between host and virus (*Unckless, 2011*). We also examined DiNV-infected *D. falleni* (a close relative of *D. innubila* with non-overlapping geographic range, collected in Athens, Georgia) as an outgroup. We sequenced 36 *D. azteca* and 56 *D. falleni*. Both viral types are present in both additional species, but the High type is rare in *D. azteca* (*Figure 6B*). The High type has a significantly higher titer than the Low type in both cases (*Figure 6A*, *D. azteca* GLM t-value = 6.71, p-value=0.0056, *D. falleni* GLM t-value = 8.12, p-value=0.000371). Viral titer is not significantly different across species for either High or Low type (*Figure 6A*, GLM t-value = −1.351, p-value=0.179). We also found the *D. azteca* samples cluster with CH *D. innubila* samples and contain the CH background SNPs (*Figure 5—figure supplement 1C*), suggesting no differentiation in the virus infecting different species. *D. falleni* DiNV, on the other hand, clusters completely separately from the other samples, likely due to its geographic separation. However, the *D. falleni* samples still have a derived cluster of High type virus, suggesting a fourth independent evolution of the High type in Georgia. Despite the lack of divergence between viruses infecting the two species in Arizona, a lower proportion of the *D. azteca* population is infected with DiNV, and the High Type is less common than the Low type DiNV (*Figure 6B*). Perhaps, even though the relative differences in titer are preserved between the two species, the Low Type is favored in *D. azteca* because this reduced virulence leads to a greater basic reproductive rate for the virus in *D. azteca*. Thus, the two types of the virus may be maintained in both host

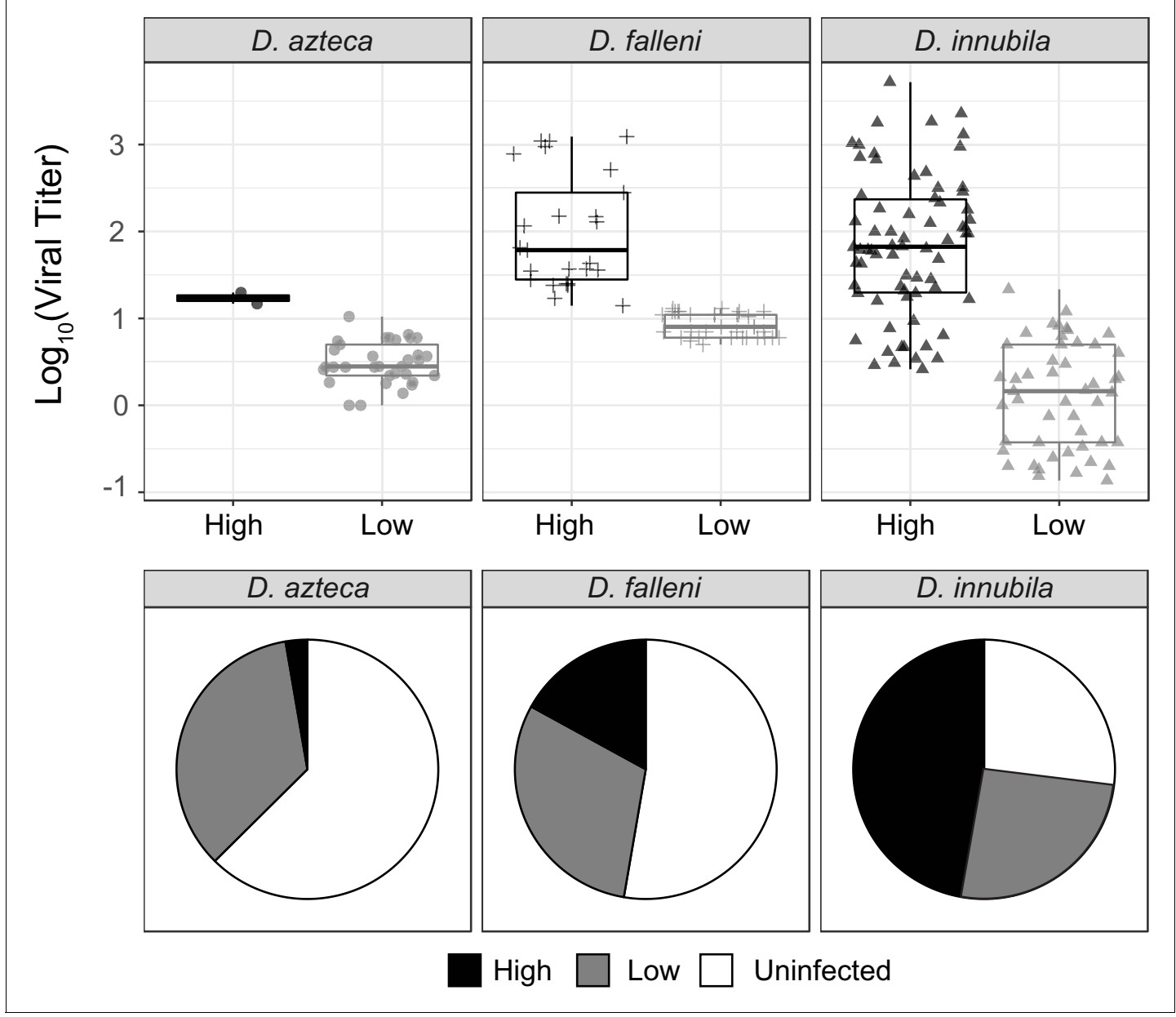

**Figure 6.** Titer and frequency of different DiNV types infecting different species. (**A**) Viral titer for CH samples of *D. azteca, D. falleni* and *D. innubila* infected with High and Low type DiNV. The *D.innubila* data presented here is a reconstruction of *Figure 1B*. (**B**) Proportion of *D. azteca* 2017, *D. falleni* and *D. innubila* 2017 CH population infected with High and Low type DiNV.

species because though they have become specialized to maximize fitness in one host, transmission between host species could lead to their continued presence in both hosts.

We repeated our association study in *D. azteca, D. falleni,* and each *D. innubila* population separately. In all cases we found the 11 High type SNPs are associated with higher viral titer (GLM t-value >4.28, p-value>0.0001 in all cases). This was not the case for the *Helicase-2* SNP (despite its presence in *D. falleni*) nor any other SNPs shared between populations. After controlling for the High type, we found no other significant DiNV SNPs in *D. azteca* associated with viral titer. For DiNV infecting *D. falleni*, we found 478 significantly associated SNPs (FDR-corrected p-value<0.01), though none of them with as large an effect as the High type associated SNPs.

# Discussion

Viruses are constantly evolving not just to propagate within a host, but also to optimize their infection across hosts. This optimization involves the relationship between their ability to infect an individual and to transmit to others, tempered by the pathogenic effects on infected hosts caused by viral activity (*May and Nowak, 1995*; *Lipsitch et al., 1996*; *Alizon and van Baalen, 2008*). Since the host is also evolving in response to the virus, an evolutionary arms-race often ensues (*Dawkins and Krebs, 1979*; *Kaltz and Shykoff, 1998*; *Daugherty and Malik, 2012*). DNA viruses have large genomes and often recombination, placing them as a somewhat transitionary pathogen between RNA viruses, bacteria and eukaryotic pathogens and parasites. Here, to work towards expanding our understanding of the co-evolution of viruses and their hosts, we examine the population dynamics of *Drosophila* innubila Nudivirus (DiNV), a DNA virus infecting *D. innubila* (*Unckless, 2011*). Within our set of viral samples, we found two DiNV types which differ by 11 SNPs (*Figure 1*, named High and Low types). One haplotype (the High type) is associated with higher viral titer, likely due to an increased manipulation of the host-immune system and increased expression of viral factors (*Figure 2*).

The derived High type has likely recurrently evolved in each population since the last glacial maximum (~10,000 years ago). The two types appear to not co-infect individuals, and mutations appear in a similar order as if navigating an epistatic fitness landscape (*Dobzhansky, 1937*; *Kondrashov et al., 2002*; *Gavrilets, 2004*). Finally, despite the higher titer and transmission rate of the High type compared to the Low type, we found the two types in *all* sampled populations. Thus, a fundamental question is: why are there two haplotypes? Possible explanations are (a) that the two haplotypes are neutral and coexist due to genetic drift, (b) we caught High type amid a selective sweep, or (c) the two haplotypes are adaptively maintained due to a tradeoff.

Given the immense differences in titer and survival between High and Low types and the rest of the preponderance of evidence presented above, we find it implausible that the two haplotypes are associated with equal viral fitness and therefore evolving under a model of genetic drift (*Gillespie, 2004*).

It is possible we caught the High type DiNV in the middle of a selective sweep in each population. Differing basic reproduction numbers ($R_0$) would result in changes in the ratio of types over time (as seen between 2001 and 2017, *Figure 1D*). As we only have two time points, we could be witnessing a selective sweep of the High type spreading to fixation (*Nielsen, 2005*), with recombination causing the observed differences in the background. As we found the High type appears to have evolved recurrently, it would be unlikely that we have caught intermediate sweeps in all four populations sampled (*Figure 1D*). Using the frequency of the High type between 2001 and 2017 CH samples, we can calculate the selection coefficient for the High type if increasing at an exponential rate (which is likely if mid-selective sweep given the intermediate frequency at both time points). *D. innubila* is active during the monsoon season if Arizona (late July to early September, 10 weeks with a generation time of 2–3 weeks) (*Patterson and Stone, 1949*). If we assume five generations per year (the minimum number of viral generations per year, limited by the host generation time) and an increase among infected individuals from 52% on 2001 to 71% in 2017, the selection coefficient = 0.0055, which suggests the High type would take ~350 years maximum to fix in a population once it has arisen (from a frequency $1/N_e$ to one in the viral population). Given the coalescence time of the two types is close to the expansion time of the two viruses (2–30 thousand years), this does not fit with our results, suggesting the two types are more likely to be adaptively maintained and not amid a selective sweep. Further, if the High type was sweeping, it would be remarkable for us to catch these sweeps occurring in all four populations sampled, given the lack of gene flow between populations, our estimated time to fixation of 350 years, and estimated initial infection ~30 thousand years ago.

Finally, two viral types could be maintained because of a tradeoff between the two virus types. We can imagine several scenarios for such a tradeoff. First, within a single population of genetically identical hosts, two viral types could coexist if they had equal basic reproduction numbers ($R_0$) (*Nowak and May, 1994*; *May and Nowak, 1995*). The basic reproduction number in a simple epidemiology model is the ratio of the transmission rate of the pathogen to its virulence. So, while we've observed that virulence is higher for the High type virus compared to the Low type, it may be also true that transmission is also higher in the High type (inferred from *Figure 1D*) and those differences are perfectly balanced creating equal $R_0$. We find this unlikely but a useful starting point for

considering the adaptive maintenance of the two types. A more likely second scenario is one where in a single population of a given species, different host genotypes favor different viral type and that the interaction between optimal $R_0$ among the viral genotypes and susceptibility of the host genotypes creates a stable equilibrium for both host genotypes and viral genotypes (*Anderson and May, 1981*). These host genotypes could also be different host species. In fact, in the Chiricahuas, *D. innubila* was more likely to be infected with the High type virus and *D. azteca* was more likely to be infected with the Low type virus (*Figure 6*) even though the two species are sympatric *and* the differences in viral titer between the High and Low types were qualitatively similar. Therefore, $R_0$ might be higher for the High type in *D. innubila* and higher for the Low type in *D. azteca* with some transmission between host species. These community dynamics could maintain both viruses in both species. The second broad category of tradeoffs involves environmental heterogeneity, which could also create the conditions for the maintenance of both viral types. Furthermore, differing environmental conditions (such as differences in population densities) could affect the frequencies in each location (*Anderson and May, 1981*). Finally, different viral types could also preferentially infect different tissues, and have different titers due to limitations of the given infected tissues. Previous work in nudiviruses identified nudivurses that infect different tissues including the gut, fat body and ovaries (*Jackson et al., 2005*; *Burand et al., 2012*; *Palmer et al., 2018*). Two DiNV strains infecting different tissues may also help explain the lack of recombination between High and Low types, as viral particles cannot exchange information if they segregate to different tissues. However, this would ignore the fact that we found no evidence of coinfection and viruses infecting two tissues could more easily coinfect.

It is possible that the High type has not recurrently evolved in each population and instead the two viral types are maintained by selection in the face of ongoing gene conversion. However, this would not explain the shared background variants which are fixed in the High type and population exclusive, or the evidence of recurrent evolution in the assembled phylogeny (*Figure 5*). It is entirely possible however that within each population, selection is maintaining the High type due to the fitness deficits of intermediate types, while gene conversion erodes the association of the eleven haplotype SNPs, which is possible given the recombination events found around the significantly associated SNPs (*Figure 3—figure supplement 1*).

Studies of different viruses have found that they can undergo speciation-like events through the accumulation of genetic incompatibilities (*Matsubara and Otsuji, 1978*; *Rokyta and Wichman, 2009*; *Meyer et al., 2016*). In these cases, the two viral types cannot recombine and produce viable viral progeny. This may be occurring in DiNV if infection by one viral type limits the ability of the other type to infect the same cell – akin to prezygotic reproductive isolation (*Coyne and Orr, 2004*). Alternatively, coinfection could happen but generate inviable recombinant particles (*Meyer et al., 2016*) – akin to postzygotic reproductive isolation. For example, the fourth mutation might only be favored when in the presence of the derived allele of the third mutation because the third mutation causes a conformational change in protein A that allows the conformational change in protein B caused by the fourth mutation to produce infective viral particles (*Orr, 1995*). The fact that *19K* is known to form a complex with other PIF proteins and other membrane proteins suggests it could play the central role in the formation of this incompatibility system (*Wang et al., 2007*; *Rohrmann, 2013*). If incompatibilities are present between types, they are likely in rapidly evolving vital nudivirus genes (*Hill and Unckless, 2017*; *Hill and Unckless, 2018*). As well as being rapidly evolving across baculoviruses and nudivurses, we found these genes also have high rates of adaptive evolution (*Figure 4*) and are associated with SNPs that distinguish the High and Low DiNV strains (*Figure 1*), further supporting the idea that the few genes are key to nudivirus replication are targets of host-suppression and locked in arms-races across the entire nudivirus/baculovirus phylogeny (*Hill and Unckless, 2017*).

DNA viruses such as DiNV have complicated replication cycles and large genomes. This makes them a sort of evolutionary intermediate between RNA viruses (small genomes, high mutation rates) and eukaryotes (large genomes, low mutation rates) and tangential to bacteria and archaea (intermediate genomes, low recombination rates) (*Rohrmann, 2013*). However, adaptation appears to occur through changes in a few key proteins (*Hill and Unckless, 2017*; *Hill and Unckless, 2018*). Here we found the evolution of two competing viral types that differ in variants near these few key genes, which cause the strains to differ in virulence and titer. Overall our results suggest that the high mutation rates and extremely high levels of selection can result in the repeated and convergent evolution

of novel host-virus interactions. Additionally, we found that these host-virus interactions for large DNA viruses can be much more complicated than previous models suggest (*Dolan et al., 2018*; *Feder et al., 2019*).

## Materials and methods

### Fly collection, DNA isolation, and sequencing

In this study, we used previously collected and sequenced *D. innubila* (*Hill and Unckless, 2020*). Briefly we collected these flies across the four mountainous locations in Arizona between the 22nd of August and the 11th of September 2017. Specifically, we collected at the Southwest research station in the Chiricahua mountains (~5400 feet elevation, 31.871 latitude −109.237 longitude), Prescott National Forest (~7900 feet elevation, 34.540 latitude −112.469 longitude), Madera Canyon in the Santa Rita mountains (~4900 feet elevation, 31.729 latitude −110.881 longitude) and Miller Peak in the Huachuca mountains (~5900 feet elevation, 31.632 latitude −110.340 longitude). Baits consisted of store-bought white button mushrooms (*Agaricus bisporus*) placed in large piles about 30 cm in diameter, at least five baits per location. We used a sweep net to collect flies over the baits in either the early morning or late afternoon between one and three days after the bait was set. Flies were sorted by sex and species at the University of Arizona and were flash frozen at −80°C before being shipped on dry ice to the University of Kansas in Lawrence, KS. During these collections we also obtained *D. azteca* which we also sorted by species and sex and flash frozen. *D. falleni* were collected using a similar method in the Smoky Mountains (~6600 feet elevation) in Georgia in 2017 by Kelly Dyer, these flies were then sorted at the University of Georgia in Athens GA and shipped on dry ice to the University of Kansas in Lawrence, KS.

For collected CH *D. innubila, D. falleni* and *D. azteca*, we attempted to assess the frequency of DiNV infection using PCR, looking for amplification of the viral gene *p47*. Using primers from *Unckless, 2011*, P47F: 5′–TGAAACCAGAATGACATATATAACGC and P47R: 5′–TCGGTTTCTCAA TTAACTTGATAGC. We used the following conditions: 95°C 30 s, 55°C 30 s, 72°C 60 s per cycle for 35 cycles. We chose the cutoff for viral infection based these PCR infection results. We compared the PCR positives with the fold coverage in the genome and found that fly samples with 10x coverage of the viral genome for 90% of the viral genome also had PCR positive results, so we chose this as our cutoff.

We sorted 343 *D. innubila* flies, 60 DiNV positive *D. falleni* and 40 DiNV positive *D. azteca* which we then homogenized and used to extract DNA using the Qiagen Gentra Puregene Tissue kit (USA Qiagen Inc, Germantown, MD, USA). We prepared a genomic DNA library of these 343 DNA samples using a modified version of the Nextera DNA library prep kit (~350 bp insert size, Illumina Inc, San Diego, CA, USA) meant to conserve reagents. We sequenced the *D. innubila* libraries on two lanes of an Illumina HiSeq 4000 run (150 bp paired-end) (Data to be deposited in the SRA). We sequenced the *D. falleni* and *D. azteca* libraries on a separate run of a lane of an Illumina HiSeq 4000 (150bp paired-end).

For 80 male *Drosophila innubila* collected in 2018 (indicated in *Supplementary file 1* - Table 2), we split the sample homogenate in half, isolated DNA from half as described above and isolating RNA using the Direct-zol RNA Microprep protocol (R2061, ZymoResearch, Irvine, CA, USA). We then polyA-selected on these samples to isolate mRNA and prepared a cDNA library for each of these 80 RNA samples using a modified version of the Nextera TruSeq library prep kit meant to conserve reagents and sequenced these samples on a NovaSeq NS6K SP 100SE (100 bp single end). We also sequenced DNA for these samples, with DNA isolated and prepared as above, also sequenced on a NovaSeq NS6K SP 100SE (100 bp single end).

### Sample filtering, mapping, and alignment

Following sequencing, we removed primer and adapter sequences using cutadapt (*Martin, 2011*) and Scythe (*Buffalo, 2018*) and trimmed all data using Sickle (-t sanger -q 20 l 50) (*Joshi and Fass, 2011*). We masked the *D. innubila* reference genome (*Hill et al., 2019*), using *D. innubila* TE sequences and RepeatMasker (*Smit and Hubley, 2008*; *Smit and Hubley, 2013*). We then mapped short reads to the masked genome and the *Drosophila* innubila Nudivirus genome (DiNV) (*Hill and Unckless, 2018*) using BWA MEM (*Li and Durbin, 2009*) and sorted using SAMtools (*Li et al.,*

*2009*). Following this we added read groups, marked and removed sequencing and optical duplicates, and realigned around indels in each mapped BAM file using GATK and *Picard, 2020* (http://broadinstitute.github.io/picard; *McKenna et al., 2010*; *DePristo et al., 2011*). We considered lines to be infected with DiNV if at least 95% of the viral genome is covered to at least 10-fold coverage. This most significantly overlaps with our CH *D. innubila* PCR results ($\chi^2$ = 71.791, p-value=2.392e-17). Additionally, index-switching at a rate of 0.0001 from the highest titer sample to an uninfected sample would still leave the uninfected with ~5.2-fold coverage of the viral genome, and so still considered as uninfected.

We then filtered for low coverage and mis-identified species by removing individuals with low coverage of the *D. innubila* genome (less than 5-fold coverage for 80% of the non-repetitive genome), and individuals we suspected of being misidentified as *D. innubila* following collection. This left us with 318 *D. innubila* wild flies with at least 5-fold coverage across at least 80% of the euchromatic genome, of which 254 are infected with DiNV (*Supplementary file 1 - Table 1*). We also checked for read pairs which were split mapped between the DiNV genome and the *D. innubila* genome using SAMtools.

For *D. falleni* we used a previously generated *D. innubila* genome with *D. falleni* variants inserted (*Hill et al., 2019*). We masked the genome with Repeatmasker (*Smit and Hubley, 2013*) and mapped short reads to the masked genome, the repeat sequences and the DiNV genome using BWA MEM and SAMtools (*Li and Durbin, 2009*; *Li et al., 2009*). Then, as with *D. innubila* we filtered for low coverage and mis-identified species by removing individuals with low coverage (less than fivefold coverage for 80% of the non-repetitive genome) leaving us with 56 *D. falleni* samples infected with DiNV.

For *D. azteca*, we downloaded the genome from NCBI (Accession: GCA_005876895.1) which we then called repeats from with RepeatModeler (*Smit and Hubley, 2008*). We masked the genome with Repeatmasker (*Smit and Hubley, 2013*) and mapped short reads to the masked genome, the repeat sequences and the DiNV genome using BWA MEM and SAMtools (*Li and Durbin, 2009*; *Li et al., 2009*). As with *D. innubila* we then filtered for low coverage and mis-identified species by removing individuals with low coverage of the *D. azteca* genome (less than 5-fold coverage for 80% of the non-repetitive genome), which left us with 37 *D. azteca* samples infected with DiNV. We then called DiNV variation using LoFreq (*Wilm et al., 2012*).

## Calling nucleotide polymorphisms across the population samples

For the 318 sequenced samples with reasonable coverage, for host polymorphism, we used the previously generated multiple strain VCF file, generated using a standard GATK HaplotypeCaller/BCFTools pipeline. We used LoFreq (*Wilm et al., 2012*) to call polymorphic viral SNPs within each of the 254 DiNV-infected samples, following filtering using BCFtools to remove sites below a quality of 950 and a frequency less than 5%. We then merged each VCF to create a multiple strain VCF file, containing 5,283 SNPs in the DiNV genome. The LoFreq VCF (*Wilm et al., 2012*) output contains estimates of the frequency of each SNP in DiNV in each sample, to confirm these frequencies, in SAMtools (*Li et al., 2009*) we generated mPileups for each sample and for SNPs of interest (related to viral titer), we counted the number of each nucleotide to confirm the estimated frequencies of these nucleotides at each position in each sample. To confirm that there are no coinfections of types, we also subsampled samples and randomly merged Low and High type samples and again generated mPileup files, for SNPs of interest we again counted the number of each nucleotide at each position and confirmed these matched our expected counts in the merged files. We then compared these artificial coinfections to actual samples to confirm the presence or absence of coinfections, finding no samples consistent with coinfections. We then used SNPeff to identify the annotation of each SNP and label synonymous and non-synonymous (*Cingolani et al., 2012*). We extracted the synonymous site frequency spectrum to estimate the effective population size backwards in time using StairwayPlot (*Liu and Fu, 2015*).

Using the viral VCF and the genetics r package we calculated $r^2$ measure of linkage disequilibrium. We visualized the linkage between the eleven focal SNPs, and between the eleven focal SNPs and 1000 random SNPs using LDheatmap (*Shin et al., 2006*). We also sorted $r^2$ scores by the types of SNPs the measure is between, if it is between the focal type SNPs, the SNPs found on each populations High type background, and other SNPs.

For 100,000 permutations, we randomized the viral titer associated with each strain. For each permutation, we binned individuals into high and low artificial haplotypes based on the allele for the SNP at 126118 (the most significantly associated SNP) and the 10 SNP most strongly linked to this SNP. Finally, we found the difference between the two artificial bins and compared this to the difference between the viral titer of the two haplotypes. We then counted the proportion of the 100,000 permutations with a difference lower than the true difference.

## Identifying signatures of adaptive evolution using McDonald-Kreitman-based tests

We filtered the total viral VCF with annotations by SNPeff and retained only non-synonymous (replacement) or synonymous (silent) SNPs. We also mapped reads from Kallithea virus (dS ~ 0.133) and Orcytes rhinoceros Nudivirus (OrNV, dS ~ 0.279) to the DiNV genome to identify the ancestral state at each site and polarize SNPs to specific branches (*Hill and Unckless, 2018*). Specifically, we sought to determine sites which are derived polymorphic in DiNV and which are substitutions fixed on the DiNV branch of the phylogeny. After removing singletons, we used the raw counts of fixed and polymorphic silent and replacement sites per gene to estimate McDonald-Kreitman-based statistics, specifically direction of selection (DoS) (*McDonald and Kreitman, 1991*; *Smith and Eyre-Walker, 2002*; *Stoletzki and Eyre-Walker, 2011*). We also used these values in SnIPRE (*Eilertson et al., 2012*), which reframes McDonald-Kreitman based statistics as a linear model, taking into account the total number of non-synonymous and synonymous mutations occurring in user defined categories to predict the expected number of these substitutions and calculate a selection effect relative to the observed and expected number of mutations (*Eilertson et al., 2012*). We calculated the SnIPRE selection effect for each gene using the total number of mutations on the chromosome of the focal gene.

For the host, we repeated this process using the SNPeff annotated VCF in the SnIPRE pipeline to identify signatures of adaptation in the host genome.

We then calculated the difference in each statistic between each gene and the median of all other viral genes, to identify how much that gene deferred from the genomic background. We repeated this for the host genome, limiting the comparison to nearby genes (within 100kbp on the same chromosome).

## Identifying differentially expressed genes between DiNV-infected and uninfected *Drosophila* innubila

For 100 male *Drosophila innubila* collected in 2018 (indicated in *Supplementary file 1* - Table 2), we homogenized each fly separately in 100 µL of PBS. We then split the sample homogenate in half, isolated DNA from half as described above and isolating RNA using the Direct-zol RNA Microprep protocol (R2061). Using the isolated DNA, we tested each sample for DiNV using PCR for *P47* as described previously, using 40 DiNV-infected samples and 40 uninfected samples. We then prepared a cDNA library for each of these 80 RNA samples using a modified version of the Nextera TruSeq library prep kit meant to conserve reagents and sequenced these samples on a NovaSeq NS6K SP 100SE (100 bp single end). We also sequenced DNA for these samples, with DNA isolated and prepared as above, also sequenced on a NovaSeq NS6K SP 100SE (100 bp single end). The DNA sequenced here was mapped as described above, with variation called as described above for other DNA samples.

Following trimming and filtering the data as described in the methods, we mapped all mRNA sequencing data to a database of rRNA (*Quast et al., 2013*) to remove rRNA contaminants. Then we mapped the short read data to the masked *D. innubila* genome and DiNV genome using GSNAP (-N 1 -o sam) (*Wu and Nacu, 2010*). We estimated counts of reads uniquely mapped to *D. innubila* or DiNV genes using HTSEQ (*Anders et al., 2015*) for each sample. Using EdgeR (*Robinson et al., 2010*) we calculated the counts per million (CPM) of each gene in each sample and counted the number of samples with CPM > 1 for each gene. We find that over 70.3% of genes have a CPM > 1 in at least 70 samples. For the remaining genes, we find these genes are expressed in all samples of a subset of the strains (e.g. DiNV uninfected, DiNV-infected, DiNV-high infected, DiNV-low infected). This supports the validity of the annotation of *D. innubila*, given most genes are expressed in some

manner, and suggests our RNA sequencing samples show expression results consistent with the original annotation of the *D. innubila* genome.

We attempted to improve the annotation of the *D. innubila* genome to find genes expressed only under infection. We extracted reads that mapped to unannotated portions of the genome and combined these for uninfected samples, samples infected with High type DiNV and samples infected with Low type DiNV as three separate samples. We then generated a de novo assembly for each of these three groups using Trinity and Velvet (*Schulz et al., 2012*; *Haas et al., 2013*). We then remapped these assemblies to the genome to identify other transcripts and found the consensus of these two for each sample. Using the Cufflinks pipeline (*Ghosh and Chan, 2016*), we mapped reads to the *D. innubila* genome and counted the number of reads mapping to each of these putative novel transcript regions, identifying 15,676 regions of at least 100 bp, with at least one read mapping in at least one sample. Of these, 717 putative genic regions have at least 1 CPM in all 80 samples, or in all samples of one group (DiNV uninfected, DiNV-infected, DiNV-low infected, DiNV-high infected). We next attempted to identify if any of these genes are differentially expressed between types, specifically between uninfected strains and DiNV-infected strains, and between Low-type infected and High-type infected strains. Using a matrix of CPM for each putative transcript region in each sample, we calculated the extent of differential expression between each type using EdgeR (*Robinson et al., 2010*), after removing regions that are under expressed, normalizing data and estimating the dispersion of expression. We find that 26 putative genes are differentially expressed between infected and uninfected types, and 69 putative genes are differentially expressed between High and Low types. We took these regions and identified any homology to *D. virilis* transcripts using blastn (*Altschul et al., 1990*). We find annotations for 37 putative genes are either expressed in all samples, or differentially expressed between samples. Of the 14 putative genes expressed in all samples, nine have the closest blast hit to an rRNA gene, and five have hits to unknown genes. For 23 differentially expressed putative genes with blast hits, three genes are like antimicrobial peptides (*IM1, IM14, IM3*), these genes are significantly downregulated upon infection, like other Toll-regulated AMPs, and have significantly lower expression in strains infected with High type DiNV compared to Low types. The remaining 20 genes all have similarity to genes associated with cell cycle regulation, actin regulation and tumor suppression genes.

## Identifying genes associated with viral titer in *Drosophila* innubila

As the logarithm of viral titer was normally distributed (Shapiro-Wilk test W = 0.05413, p-value=0.342), we used PLINK (*Purcell et al., 2007*) to associate nucleotide polymorphism to logarithm of viral titer in infected samples. We first generated a relationship matrix for the viral samples using PLINK. This kinship matrix allows us control for pairs of individuals who, on average, have similar alleles and which, on average, have dissimilar values. We then fit a linear model in PLINK including population, sex, *Wolbachia* presence, the date of collection and the distance matrix for relationship of each sample (inferred using PLINK, shown as relationship[strain]).

Before performing the association study, we removed host SNPs found in fewer than five samples and merged SNPs in perfect linkage within 10kbp of each other, filtering down from 5283 viral polymorphisms to 1403 viral polymorphisms. We then identified associations between the logarithm of viral titer and the frequency of the viral polymorphism in each individual sample, resulting in the following model:

$$Log_{10}(viral\ titre) \sim SNP + hs + w + p + dc + (SNP * hs) + \\ (SNP * p) + (SNP * w) + relationship[strain]$$

Where *hs* = host sex, p=population of collection, *w* = *Wolbachia* presence, *dc* = date collected, relationship[strain]=distance matrix. Following model fitting, performed a posthoc analysis to determine if any clumped SNPs were significant and if they should be assessed separately, we also identified covariates which seemed to show little or no effect on viral titer (p-value>0.1) using an ANOVA in R (*R Development Core Team, 2013*), and removed these, refitting the model. This was done step-wise, resulting in the following model:

$$Log_{10}(viral\ titre) \sim SNP + hs + p + (SNP * hs) + relationship[strain]$$

Following this we also performed an association study using PLINK (*Purcell et al., 2007*) in the

host, using previously called host variation, and considering viral haplotype as an additional covariate. Before we performed this analysis, we clumped SNPs that are strongly linked ($r^2$ >0.99) within 10kbp of each other.

$$Log_{10}(viral\ titre) \sim SNP + hs + p + (SNP * hs) + vh + relationship[strain]$$

Whereas previous, and with *vh* = viral haplotype. Using GOrilla (*Eden et al., 2009*), we found no gene categories enriched in the significant SNPs (*Figure 4—figure supplement 3*).

We repeated this analysis for DiNV variants in *D. azteca* and *D. falleni* separately. We performed the association study twice, first using the original model, then including viral haplotype as an additional covariate.

Because we are performing multiple correlated tests, we next determined the genome-wide significance threshold for the association between a SNP and the viral titer by permutation. We permuted the viral titer information randomly across samples and repeated the genome-wide association studies as described above. We recorded the minimum *p*-value across the genome from 1000 permutations to generate a null distribution. To identify if the SNPs involved in the High type are more likely to appear together than expected, we again used a permutation test to identify the frequency that these combinations occur across 1000 permutations, where a P of 0.01 = 7.987812e-10.

## Estimating viral titer using qPCR

Following the identification of the viral haplotype associated with viral titer we sought to determine the effect of viral haplotypes in actual infections. For 20 samples with fly homogenate, we determine the viral titer and haplotype following filtration with a 0.22 µM filter.

We performed qPCR for the viral gene *p47* (Forward 5-TCGTGCCGCTAAGCATATAG-3, Reverse 5-AAAGCTACATCTGTGCGAGG-3) on 1 µL of fly filtrate per sample and compared the estimated Cq values across three replicates to estimated viral copy number to confirm viral concentration (protocol: 2 min at 95˚C, 40 cycles of 95˚C for 30 s and 59˚C for 20 s, followed by 2 min at 72˚C). Following this we diluted samples to similar Cq values, relative to the sample with the highest Cq value. We confirmed this by repeating qPCR with *p47* primers of 1 µL of each sample.

For each filtrate sample we performed infections on 30 *D. innubila* males 4–5 days following emergence using pricks with sterile needles dipped in viral filtrate. We recorded survival of each fly each day and removed dead flies. Finally, we took samples 1, 3, and 5 days post infection and measured viral copies of *p47* relative to *tpi* at each time point.

## Phylogeography of DiNV infection and the evolution of the different viral haplotypes

For each DiNV-infected *D. innubila* sample, we reconstructed the consensus DiNV genome infecting them using GATK AlternateReferenceMaker and the VCF generated for each strain (*McKenna et al., 2010*; *DePristo et al., 2011*). We used BEAST2 with the SkyLine package (*Stadler et al., 2013*) to build the phylogeny of DiNV genomes using 100 million iterations with a burn in of 1 million, sampling every 1000 trees (*Bouckaert et al., 2014*). After generating a set of phylogenies in BEAST2, we checked that convergence had occurred across the samples using Tracer and generated a final consensus phylogeny using TreeAnnotator (*Bouckaert et al., 2014*). For each High type SNP, we reconstructed the evolution of the SNP as character states across the BEAST2 phylogenies to infer the SNP appearance order. We used the all different rates (ARD) discrete model in APE (*Paradis et al., 2004*) on 1000 randomly sampled converged BEAST2 phylogenies to infer SNP states at each branch and calculated the bootstrap support for the order of appearance for each mutation in each population. To independently identify recurrent mutations across the DiNV phylogeny, we also used TreeTime (*Sagulenko et al., 2018*). Finally, we created a matrix of SNPs present in at least two viral samples and used this matrix in a principal component analysis in R (*R Development Core Team, 2013*), labeling each sample by their viral type in the PCA (based on the presence or absence of the significant *19K* SNP).

## Simulating the evolution of the high and low viral haplotypes

We sought to simulate the infection of DiNV in *D. innubila* when considering the evolution of a high titer viral haplotype, specifically if two viral types can be maintained against each other at stable frequencies, and if the high viral haplotype with five shared mutations could evolve recurrently in the given time period given realistic parameters. We used the R package DeSolve (*Soetaert et al., 2010*) to simulate infection dynamics in an SI model. We did not include the resistance class, under the assumption that flies won't live long enough to clear the infection. Therefore, the proportion of population infected per generation is as follows:

$$\frac{d_{Infected}}{d_{Time}} = (Susceptible * Infected * \beta) - (Infected * \gamma)$$

Where $\beta$ = infection parameter (e.g. the increased likelihood an infected individual spreading its infection before dying) and $\gamma$ = virulence parameter (e.g. the increased likelihood an infected individual has of dying before it can spread its infection). Infected = The proportion of the population infected with DiNV.

Susceptible = the proportion of the uninfected population.

We attempted to assess if the 'high titer' viral haplotype could possibly evolve recurrently in each population in the time scale seen in our findings. We again used the SI model, this time discrete with population sizes set to 1 million individuals, based on *StairwayPlot* estimates (Liu and Fu 2015), starting with 1 infected individual ($1/N_e$). We reasoned that if the wait time between beneficial mutations is $1/(2N_e*u*s)$, and increasing viral titer also increases $N_e$, then the increased titer also decreases the wait time between the appearance of mutations. For each infected individual in the population, we tracked the viral titer and also recorded the presence of absence of five mutations, with each mutation increasing the titer of infection, but each further mutation having successively smaller increases in viral titer ($titer^{\sqrt{no.\ muts}}$), representing the epistatic interaction of high titer associated mutations seen in the viral haplotype. Viral titer did not increase if a mutation occurs before the preceding mutation was present occurred, to matching the order of appearance of mutations seen in DiNV. In all simulations, we assumed 5 viral generations per year, based on the assumption of 5 host generations per year being the minimum number of viral generations (likely much higher), and so useful to conservatively estimate the amount of time required for the recurrent evolution of the High type.

In the simulations, we multiplied the infection parameter, mutation parameter and virulence parameter by viral titer, under the assumption that viral titer increases both infection and death rate, and the mutation rate is per viral particle (*Maeda et al., 1993*; *Kamita et al., 2003*). We considered a per site mutation rate of $10^{-6}$ mutations per generation, based on estimated baculovirus mutation rate (*Rohrmann, 2013*; *Chateigner et al., 2015*), with a specific mutation rate for each of the high titer variants of 3.3e-7 (1e-6 mutations per base per generation * 1/3 for the correct mutation) * viral titer, where each mutation occurs independently (so all five could arise in one generation at a rate of $(3.3e-7)^5 = 4.12e-33$). We then simulated populations in replicate 1000 times for 100,000 generations with a starting infection frequency of 10% for the 'low titer' haplotype, recording the frequency of the virus in a population, the frequency of the haplotype and the time that each 'high titer' mutation reaches high enough frequency to escape stochastic behavior and behave deterministically under selection (*Gillespie, 2004*). We also factored in recombination between each variant site a rate of 2% per site-window per generation (*Kamita et al., 2003*). Specifically, we randomly paired viral genomes and in 2% of pairs we randomly recombined the variant combinations. We did not increase recombination rate as titer increased as this was not observed in nature (*Kamita et al., 2003*).

To estimate the possible selection coefficient for DiNV in the CH population, we assumed an exponential distribution and five host generations per year (80 host generations between 2001 and 2017). We then solved the following equation:

$$P_{2017} = P_{2001} * (1+s)^t$$

Where $P_{2017}$ = the frequency of the High type among viral samples in 2017 (71%), $P_{2001}$ = the frequency of the High type among viral samples in 2001 (52%), s = the selection coefficient and t = the number of generations (80). We then used this estimated selection coefficient in the same equation to find the number of generations (t) to go from $1/2N_e s$ (0.0000714, assuming an $N_e$ of 1000000) to fixation (0.99):

$$0.99 = 0.0000714 * (1 + 0.0055)^t$$

## Experimental infections of *Drosophila* innubila with DiNV

We chose *D. innubila* samples infected with DiNV and with sequenced genomes, four infected with the High type DiNV and four infected with the Low type. For these samples we estimated their viral copy number per host genome as described previously. We used qPCR on *p47* and *tpi* to find the differences in Cq values to calculate the concentration of each sample relative to the lowest concentration sample and diluted 50 µL of filtrate for each sample to match the concentration of each sample to the samples with the lowest titer (IPR07). For a separate 50 µL of the IPR01 sample, we performed 1 in 10 serial dilutions to give 45 µL of filtrate at full concentration, 1 in 10 concentration, 1 in 100 concentration and 1 in 1000 concentration. Using these sets of samples (matched titer and serial dilutions) we next performed experimental infections.

We transferred 50 *D. innubila* (of roughly equal sex ratio) to new food and let them lay eggs for 1 week, following this we collected male offspring aged 2–5 days for experimental infections.

Across four separate days in the mornings (between 9am and 11am), we infected the collected male flies with each sample. For flies in batches of 10, we performed pricks with microneedles dipped in the prepared viral filtrate. For each day we also had two control replicates of 10 flies pricked with microneedles dipped in sterile media. Following infections, we checked on each vial of 10 flies one-hour post infection and removed dead flies (likely killed by the needle instead of the virus). We also checked each vial each morning for 15 days, removing dead flies (freezing to determine the viral titer), and flipping flies to new food every 3–4 days. Checking at 10am each day, we recorded the day that each fly died, what filtrate they had been infected with, and what replicate/infection day set they belonged to. We next looked for differences in survival over time compared to sterile wound controls using a Fit proportional hazards regression model in R (*R Development Core Team, 2013*; *Kassambara et al., 2017*), considering titer, viral isolate (nested in haplotype) and replicate/vial as co-variates (day of death ~ [titer or haplotype | strain] + infection date).

For a second set of experimental infections (performed as described above, stabbed with diluted filtrate from different strains), we also removed three living flies 1 hr, 1 day, and 5 days post infection. Using qPCR, we found the difference in *p47* log-Cq and *tpi* log-Cq to estimate the viral copy number for each sample over time.

## Acknowledgements

This work was completed with helpful discussion from Justin Blumenstiel, Joanne Chapman, John Kelly, Stuart MacDonald, Andrew Mongue and Carolyn Wessinger. We would especially like to thank Maria Orive, Kelly Dyer and Paul Ginsberg for helpful feedback in the writing of the manuscript and framing of the discussion. Collections were completed with assistance from Todd Schlenke, Paul Ginsberg, Kelly Dyer, Brandon Cooper, John Jaenike and the Southwest Research Station. We thank Brittny Smith and Jenny Hackett at the KU CMADP Genome Sequencing Core (NIH Grant P20 GM103638) and K-INBRE Bioinformatics Core for assistance in genome isolation, library preparation, sequencing and computational resources. This work was supported by a K-INBRE postdoctoral grant to TH (NIH Grant P20 GM103418). This work was also funded by NIH Grants R00 GM114714 and R01 AI139154 to RLU. *D. falleni* collection was funded by NSF grant DEB-1737824. Supplementary Data 1-9 is available in the following data dryad folder: https://doi.org/10.5061/dryad.2fqz612mh. (doi:10.5061/dryad.2fqz612mh). Additional data regarding *D. innubila* population genomics is available in the following FigShare folder: https://figshare.com/projects/innubila_population_genomics/87662.

## Additional information

### Funding

| Funder | Grant reference number | Author |
|--------|------------------------|--------|
| KU CMADP | P20 GM103638 | Tom Hill<br>Robert L Unckless |

| K-INBRE | P20 GM103418 | Tom Hill |
| --- | --- | --- |
| National Institutes of Health | R00 GM114714 | Robert L Unckless |
| National Institutes of Health | R01 AI139154 | Robert L Unckless |
| National Science Foundation | DEB-1737824 | Tom Hill<br>Robert L Unckless |

The funders had no role in study design, data collection and interpretation, or the decision to submit the work for publication.

### Author contributions
Tom Hill, Conceptualization, Data curation, Software, Formal analysis, Funding acquisition, Investigation, Visualization, Methodology, Writing - original draft, Project administration, Writing - review and editing; Robert L Unckless, Conceptualization, Supervision, Validation, Methodology, Writing - original draft, Project administration, Writing - review and editing

### Author ORCIDs
Tom Hill (iD) https://orcid.org/0000-0002-4661-6391
Robert L Unckless (iD) https://orcid.org/0000-0001-8586-7137

### Decision letter and Author response
Decision letter https://doi.org/10.7554/eLife.58931.sa1
Author response https://doi.org/10.7554/eLife.58931.sa2

## Additional files

### Supplementary files
• Supplementary file 1. Next-generation sequencing information of Drosophila infected with DiNV used in this survey. Table 1: Summary of *Drosophila innubila* and *D. azteca* fly samples collected and sequenced for this study, table includes summary of coverage for X chromosome,, Muller B, other autosomes, virus and *Wolbachia.* Also contains SRA accessions for each strain. Table 2: Summary of *Drosophila* innubila fly RNA and DNA collected and sequenced for this study, including if infected with DiNV.

• Transparent reporting form

### Data availability
Sequencing data have been deposited on the NCBI SRA under the study accession: SRP187240 Genomes used in this study are available at the following accessions: *Drosophila innubila* - GCF_004354385.1 *Drosophila* innubila Nudivirus - GCF_004132165.1 *Drosophila* azteca - GCA_005876895.1.

The following datasets were generated:

| Author(s) | Year | Dataset title | Dataset URL | Database and Identifier |
| --- | --- | --- | --- | --- |
| Hill T, Unckless RL | 2020 | *Drosophila* Sky Island sequencing | https://www.ncbi.nlm.nih.gov/sra/SRX5449484 [accn] | NCBI Sequence Read Archive, SRP187240 |
| Hill T, Unckless RL | 2020 | *Drosophila* Sky Island data analysis | http://dx.doi.org/10.5061/dryad.2fqz612mh | Dryad Digital Repository, 10.5061/dryad.2fqz612mh |
| Hill T, Unckless RL | 2020 | Innubila population genomics | https://figshare.com/projects/innubila_population_genomics/87662 | figshare, innubila_population_genomics/87662 |

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
