## [Decision Letter]

**Acceptance summary:**

This study reports on a phenotypic and genetic polymorphism of DiNVirus in natural populations of *Drosophila*. The authors present a series of experiments and assessments to understand how the polymorphism evolved and what implication it has for the host and conclude that the observed polymorphism arose multiple times independently and that it is maintained in a polymorphic state. The results are very clear and convincing and provide an excellent example for the power of natural selection in shaping host-parasite interactions.

**Decision letter after peer review:**

Thank you for submitting your article "Recurrent evolution of two competing haplotypes in an insect DNA virus" for consideration by *eLife*. Your article has been reviewed by three peer reviewers, one of whom is a member of our Board of Reviewing Editors, and the evaluation has been overseen by Diethard Tautz as the Senior Editor. The following individual involved in review of your submission has agreed to reveal their identity: Sebastian Gagneux (Reviewer #3).

The reviewers have discussed the reviews with one another and the Reviewing Editor has drafted this decision to help you prepare a revised submission.

Your manuscript presents an exciting finding of the evolution and biology of a natural viral pathogen of *Drosophila*. You suggest that two distinct haplotypes of the virus evolved multiple times in isolated host populations. These two haplotypes are characterised by 11 SNPs and associated phenotypic traits (differences in viral titer). The independent evolution of such a complex trait seems rather unusual, making this nice example of evolution of host-parasite interactions.

The three reviewers of this manuscripts have a lot of praise for the study. However, they also raise a number of important points that needs to be addressed. Most important, the substantive technical points raised by reviewer 2 need careful consideration. This includes explaining and resolving the inconsistencies in the analysis and the right choice of the methods. Reviewer 1 pointed out problems with the simulation. Below are the detailed reviews. A final decision will only be possible when the problems in the analysis are solved.

Reviewer #1:

This study reports on a phenotypic and genetic polymorphism of DiNVirus in *Drosophila*. The study is very appealing as this study system is a natural system (unlike *D. melanogaster*) with a well understood ecology and biogeography. The authors go through a series of experiments and assessments to understand how the polymorphism evolved and what implication it has for the host. The main conclusion is that the observed polymorphism arose multiple times independently and that it is maintained in a polymorphic state. The mechanism for this maintenance is not entirely clear. A simulation model is used to bring some light into this puzzle. For the most part, the study is solid and well carried out. Below are a number of points that may help the authors to present their material more clearly.

The Introduction is not to the point. The Introduction moves among various aspects of host-parasite interactions without giving the reader an idea where this is going. At various places topics are raised that then later dismissed. For example: the first two paragraphs are about host response to viruses. In the third paragraph it become system specific, but now switches mainly to the DiNV system. It is not clear where one is going here. The Introduction (sixth paragraph) is summed up with a very generic phrase without much perspective on what is going to follow. At this place I still do not know what this paper will be about. A much more targeted Introduction is necessary. What are the questions? What was driving this research? What are the hypotheses?

With 11 SNPs spread across the chromosome and obligate recombination every round of replication, the proportion of viral offspring with sub-optimal multi-SNP-genotypes must be huge. How can the right genotypes be maintained? Discuss.

The authors run computer simulations (called SIR models, but they seem actually to be SI models) to support their ideas about virus evolution. The simulation regarding the accumulation of mutation is fine and gives quantitative support for presented evolutionary scenario.

A second simulation is about the competition of the two viral types. This is by far the weakest part of the manuscript. I am not convinced about the value of this simulation. The outcome can be predicted from the assumptions of the model, several of them are very speculative. Without data on transmission over the course of the infection, the simulations are not very helpful. I suggest to leave this out. It is ok to speculate about this in the Discussion, but it makes the Results part heavy and less strong. Also, the associated figure (Figure 7) is hard to understand.

The authors stress at multiple place that it is likely that the increased virulence of the high type is traded-off against transmission. The evidence for this is much weaker than the strength of these statements suggests. I strongly suggest to tone this down.

Reviewer #2:

This manuscript presents an exciting analysis of the evolution and biology of a natural viral pathogen of *Drosophila*. In outline: the authors claim that two distinct haplotypes of the virus co-occur in multiple host populations, that these 'evolved' independently (i.e. separate origins) through convergence, and that they have alternative 'life histories' (high titer and virulence, versus low titer and virulence), which may permit their coexistence.

The practical experiments and sequencing data are substantial and appear sound, and the work is likely to be of interest to a very broad host-pathogen audience. However, while some of these headline claims are well-supported, I have a number of serious concerns about the analyses and interpretation of others, and a few key methodological details are missing. The analyses would need substantial revision, or at least additional checks, before I would be convinced by the story.

1) Is there any possibility of circularity in defining the 'high' and 'low' haplotypes?

Two 'types' are defined based on 11 linked SNPs that are described as having 'high' or 'low' titer phenotypes. First, this requires a more robust approach to define 'significance' as (if there is any LD) tests are not independent, and sample sizes relative to predictive SNPs are small. Phenotypes (titers) should be permuted across genotypes (within populations) a thousand times and the analyses re-run for each permutation, then the tails of this distribution used to define significance, as commonly done for e.g. the DGRP. Second, this feels like it is dangerously circular: supposing all 11 SNPs were false positives, and arbitrary haplotypes erroneously defined based on them? Post hoc analysis of the difference between the two haplotypes could still show Figure 1B, because those differences were what drove the (erroneous) detection of the SNPs. So would Figure 1C, since that result necessarily follows from the definition of the haplotypes.

To convince me that the 'two haplotypes' interpretation is real, for each of the randomisation replicates one would need to define a 'high' and a 'low' haplotype based on the any randomisation spurious 'significant' SNPs and re-run the rest of Figure 1 to demonstrate that the separation between the real haplotypes is greater than that between those that result from permutation tests. I appreciate that this will require some computing time, but the authors already have all of the code, so it shouldn't be more than an afternoon of 'hands on' time.

Since the haplotypes overlap in phenotype space (Figure 1B) and not all have all SNPs (Figure 1C), how were intermediate haplotypes (with <11 SNPs) assigned to high or low?

2) Did the haplotypes evolve independently three or four times?

Of the 11 SNPs defining the haplotype, three are non-synonymous, five are in the UTRs of known virulence genes, and three are intergenic SNPs. Is it really credible that a specific base change has arisen and been selected for independently in each of four populations, at each of the 11 sites? i.e. always A->C being beneficial at a site, but never A->G at that site? Even for the non-coding ones? This is an extraordinary claim that would require extraordinary evidence, for which the rates of recombination and gene conversion seem at the heart of the argument.

In some places the authors appear to assume (or assert) that recombination is frequent, while in others they assume it is completely absent, and nowhere do they explicitly test for it. If it is common, then none of the tree-based analyses can be used. If it is absent, then it is very hard to explain the patterns of diversity or LD in Figure 1—figure supplement 3, and some of the simulations may be inappropriate.

The tree-based analyses seem to be the basis of the major claim that these haplotypes arose independently multiple times, and that the order of the mutations arising was similar. Any tree analysis absolutely requires the absence of recombination or gene conversion, and this needs to be explicitly tested for here (e.g. using GARD, or possibly if diversity is low by inferring the ARG using tsinfer). The failure of LD to decay with distance hints that recombination is absent, but gene conversion over very short distances could still shuffle mutations between haplotypes. However, assuming its branch lengths are in mutations, the shape of the tree in Figure 6A (many short tips below large crowns) shouts 'recombination' to me. Where did this tree come from? It is not ultrametric, so if it was inferred with BEAST this is not standard output.

If the authors do have evidence that recombination is absent, can they confirm that the other analyses (population size history from the SFS; simulations) also assumed zero recombination? Although zero recombination would make me worry even more about the p-values in the GWAS discovery of the SNPs that define the haplotypes – but permutation tests would help deal with that.

I am really confused about their view of recombination, because a couple of times they imply recombination may be common, for example by noting that recombination is required in Nudivirus replication – but then this would invalidate the tree-based analyses? Plus, mechanistic recombination is irrelevant without coinfection infection, since recombination only between identical haplotypes does not have any effect. The observation that no co-infections occur suggests recombination should be very rare – but that's not what the tree looks like. The sixth paragraph of the Discussion suggests that not enough thought has been given to the likelihood, or implications, of recombination.

Finally, if there is no recombination, then 'fully derived' haplotypes (those with all 11 SNPs) can only be as old as the youngest of the 11 SNPs, unless the SNPs arose multiple time within populations as well. Is that compatible with the patterns of diversity and the timescale? One check would be to mask those 11, re-infer the tree, and check that the clades are still monophyletic. If these arose by re-current mutations due to strong selection, they could be warping the tree – this might be akin to the problem of recurrent drug or MHC selected mutations in HIV trees, where the known sites are excluded before analysis.

If we believe the two haplotypes are real, a much more credible 'story' to me would be two distinct and potentially old haplotypes maintained by selection in the face of a low level of ongoing gene conversion (or recombination). Is there some aspect of the data that this does not fit? This seems just as good a 'story', just as interesting, and just as publishable.

3) The MK-like analyses

Although a relatively minor part of the story, the MK analysis is extremely unclear. In part this is because it doesn't appear in the Materials and methods.

i) What software was used? The supporting data implies VCFtools can do this (which surprised me) but the method it uses was not explained. Was SNIpre fitted with the original code? The text seem to imply that raw counts were used, which would not be suitable unless Ks was <0.3

ii) Although divergence is required, we're not told how (or from what species) it was estimated. If Ks>0.3, then something more than raw counts should certainly be used. If Ks>0.8, I would strongly advise against doing any sort of MK analysis.

iii) Figure 4 gives 'difference from background'. What is meant by difference from background? But if the background is a signal of strong constraint, could a positive signal here mean relaxed constraint rather than positive selection. What is the evidence that it's positive selection rather than relaxed constraint?

iv) In Figure 4—figure supplement 1 the populations are presented separately, and differences among them are interpreted as differences in positive selection. But surely the divergence number must be massive, larger than the polymorphism number, so almost all of this variation is due to differences in constraint affecting the Pn/Ps ratio. Or is the method one that uses high frequency derived SNPs as evidence of positive selection? If so, how was ancestral state identified? This is probably not possible to do reliably if Ks>0.3. We need to see some raw numbers, and a lot more detail on the methods.

Reviewer #3:

This is an interesting piece of work on the co-evolution of the *Drosophilainnubila* Nudivirus (DiNV) and its host. The authors analyzed several natural populations of *D. innubila*, some of which were partially infected with DiNV, and simultaneously characterized both the host and the infecting viral pathogen using a combination of DNA and RNA sequencing and various complementary analytical approaches. Using a GWAS approach, they discovered a viral variant with high virulence (High Type) that differed by 11 strongly linked SNPs from low virulence variants. They demonstrate that the High Type associates with a higher viral titer and increased host mortality, and also validate these findings using experimental infection assays. Using a transcriptomic approach, they show that the High Type overexpresses genes known to be linked to viral virulence, and this correlated with the under expression of host genes involved in antiviral immunity, indicating that the increased virulence of the High Type is at least partially due to the inhibition of host defense mechanisms. They further show that loci associated with differences in virulence were under strong selection for adaptation, particularly genes involved in the viral envelope and virulence proteins. Based on their reconstruction of the most likely evolutionary histories of the High Type in the different host populations, they further conclude that the High Type emerged multiple times independently, and yet, the High Type did not outcompete the Low virulence variants in these populations. This might indicate varying trade-offs between virulence and transmission in the High and Low Types that vary across these populations. Finally, they show that the same phenomenon for High Type evolution of DiNV can also be observed in other *Drosophila* species. I have just a few comments:

1) Throughout the manuscript, the authors switch back and forth between the present and past tense, which seems awkward from a stylistic point of, I'd suggest to stick to the past tense through-out.

2) The fact that the exact same 11 SNPs evolve multiple times independently in mostly the same order is interesting. The authors note that an expected alternative could be different SNPs emerging in the same genes, but they don't discuss the potential mechanism, by which the exact same SNPs seem to be preferred instead.

3) The authors found little evidence of mixed infection with both the High and Low Types and conclude potential in-compatibility. Please expand on the potential mechanisms of this.

4) Related to the above comment, the authors observe no "hybrids" of High and Low Types again, suggesting "incompatibility". Please discuss the difference between this genetic/genomic versus ecological incompatibility referred to above, as well as the potential link between these two types of incompatibilities.

5) Please rephrase the first sentence of the Discussion (what is the deference between "to better infect" and "optimizing the infection"?).

---

## [Author Response]

Reviewer #1:[…] Below are a number of points that may help the authors to present their material more clearly.The Introduction is not to the point. The Introduction moves among various aspects of host-parasite interactions without giving the reader an idea where this is going. At various places topics are raised that then later dismissed. For example: the first two paragraphs are about host response to viruses. In the third paragraph it become system specific, but now switches mainly to the DiNV system. It is not clear where one is going here. The Introduction (sixth paragraph) is summed up with a very generic phrase without much perspective on what is going to follow. At this place I still do not know what this paper will be about. A much more targeted Introduction is necessary. What are the questions? What was driving this research? What are the hypotheses?

We have rewritten the Introduction following your suggestions, thank you for the comments regarding this. We have moved sections around and rewritten parts so there is a better flow between paragraphs. We have also better provided our initial hypotheses and how these led to our findings. Specifically, we use the first paragraph to describe the host/virus coevolution and how viruses suppress the host, we then outline why people study viruses and why the DiNV system is ideal for viral study (with details of the system). Finally, we use the last paragraph to discuss the experiments we performed and a basic outline of our results.

With 11 SNPs spread across the chromosome and obligate recombination every round of replication, the proportion of viral offspring with sub-optimal multi-SNP-genotypes must be huge. How can the right genotypes be maintained? Discuss.

We have further expanded upon this in both the Results and the Discussion. In the Results we highlight that though there is obligate recombination, the chance that two strains with differing genotypes infect the same cell in an actual organism is very low, so we cannot know the actual rate or recombination, but it is likely lower than the high rate suggested and would lead to a lower proportion of intermediate strains. In the Discussion we also cover this and highlight that one possible difference between the High and Low strains is that they could infect different tissues (as is seen between other nudivirus strains). If the two types preferentially infect different tissues, then the chance of them co-infecting the same cell is impossible and would result in our observed absence of recombination between the High and Low type. We also discuss incompatibilities that could occur in suboptimal genotypes which could be the cause of their absence in our survey, or other factors which could drive the maintenance of the full haplotypes.

The authors run computer simulations (called SIR models, but they seem actually to be SI models) to support their ideas about virus evolution. The simulation regarding the accumulation of mutation is fine and gives quantitative support for presented evolutionary scenario.A second simulation is about the competition of the two viral types. This is by far the weakest part of the manuscript. I am not convinced about the value of this simulation. The outcome can be predicted from the assumptions of the model, several of them are very speculative. Without data on transmission over the course of the infection, the simulations are not very helpful. I suggest to leave this out. It is ok to speculate about this in the Discussion, but it makes the Results part heavy and less strong. Also, the associated figure (Figure 7) is hard to understand.

We have now removed the second set of simulations and associated figures regarding the possibility of virulence/transmission equilibrium, as we agree it is the weakest part of the manuscript and is unlikely, as it is an unstable equilibrium. We now discuss the idea of a virulence/transmission equilibrium in the Discussion, however, we also discuss all the possible things that could have resulted in us finding two viral types (incomplete sweep, trade-off, optimized to different hosts).

The authors stress at multiple place that it is likely that the increased virulence of the high type is traded-off against transmission. The evidence for this is much weaker than the strength of these statements suggests. I strongly suggest to tone this down.

We have rewritten our Discussion to tone this down, in our Results we describe a relationship we find between increasing titer and increasing virulence. Beyond this, in the Discussion we examine multiple possible causes for the two maintained strains. We hypothesize that the two strains could be neutral and are drifting, that the High type is partway through a sweep, that a trade-off allows them to exist alongside each other or that each virus type is optimized to different host species/genotypes (with migration between two types). Overall, we think we have removed the majority of our Results and Discussion where talk about the trade-off, as it is not the only suggested explanation.

Reviewer #2:[…] The practical experiments and sequencing data are substantial and appear sound, and the work is likely to be of interest to a very broad host-pathogen audience. However, while some of these headline claims are well-supported, I have a number of serious concerns about the analyses and interpretation of others, and a few key methodological details are missing. The analyses would need substantial revision, or at least additional checks, before I would be convinced by the story.1) Is there any possibility of circularity in defining the 'high' and 'low' haplotypes?Two 'types' are defined based on 11 linked SNPs that are described as having 'high' or 'low' titer phenotypes. First, this requires a more robust approach to define 'significance' as (if there is any LD) tests are not independent, and sample sizes relative to predictive SNPs are small. Phenotypes (titers) should be permuted across genotypes (within populations) a thousand times and the analyses re-run for each permutation, then the tails of this distribution used to define significance, as commonly done for e.g. the DGRP. Second, this feels like it is dangerously circular: supposing all 11 SNPs were false positives, and arbitrary haplotypes erroneously defined based on them? Post hoc analysis of the difference between the two haplotypes could still show Figure 1B, because those differences were what drove the (erroneous) detection of the SNPs. So would Figure 1C, since that result necessarily follows from the definition of the haplotypes.

There are two important considerations here so we will address them separately. We address the false negative problem below.

Defining significance since tests are not independent: Based on reviewer comments, we have significantly revised our approach at examining linkage disequilibrium and hope this clarifies some of the concerns. Given the signatures of LD, we did perform permutations and now include the permutation threshold in Figure 1A (slightly less stringent than the corrected P-value with a FDR of 0.01). So this was an important improvement but did not qualitatively change the result.

To convince me that the 'two haplotypes' interpretation is real, for each of the randomisation replicates one would need to define a 'high' and a 'low' haplotype based on the any randomisation spurious 'significant' SNPs and re-run the rest of Figure 1 to demonstrate that the separation between the real haplotypes is greater than that between those that result from permutation tests. I appreciate that this will require some computing time, but the authors already have all of the code, so it shouldn't be more than an afternoon of 'hands on' time.

The haplotype association problem: This is an important concern – if the first SNP is a false positive, they are all false positives. We think we have three convincing lines of evidence that these are not false positive SNPs. First, we have now performed the association study in the five populations (3 *innubila*, 1 *azteca* and 1 *falleni*) independently. Given the viral population structure among these populations (except maybe the sympatric *innubila*/*azteca*), these represent 5 independent tests and the haplotypes are significantly associated in all five populations. Thus a false negative would have to be a false positive 5 (or 4 if we don’t consider sympatric *innubila*/*azteca* as separate) times. Even if we consider our FDR threshold of 0.01 (these SNPs have lower P values), the likelihood of a false positive in 5 independent tests is 0.01^5 = 10^-10 or 1 in 10 billion (if 4 independent tests, that value drops to 10^-8 or 1 in 100 million). Second, we have followed the reviewer’s suggestions (we think) and permuted the titer associated with each strain 100,000 times, and each time binned strains by the most significant SNP and its 10 most closely linked SNPs, then find the difference in titer between two types. We find that the difference between the high and low type is significantly higher than by random chance for all SNPs, and no random combinations of alleles show the titer increase seen in Figure 1B. We have included the distribution of permuted differences between the artificial high and low types in Figure 1—figure supplement 2, with the true difference shown as a dotted line.

Since the haplotypes overlap in phenotype space (Figure 1B) and not all have all SNPs (Figure 1C), how were intermediate haplotypes (with <11 SNPs) assigned to high or low?

We have now changed the manuscript to consider intermediate strains to be a third subset of DiNV and have excluded them from analyses comparing High to Low type, including Figure 1B.

2) Did the haplotypes evolve independently three or four times?

As we have clarified in the manuscript, the haplotype has recurrently evolved three times to our knowledge in the Sky Islands, and at least once more in another location.

Of the 11 SNPs defining the haplotype, three are non-synonymous, five are in the UTRs of known virulence genes, and three are intergenic SNPs. Is it really credible that a specific base change has arisen and been selected for independently in each of four populations, at each of the 11 sites? i.e. always A->C being beneficial at a site, but never A->G at that site? Even for the non-coding ones? This is an extraordinary claim that would require extraordinary evidence, for which the rates of recombination and gene conversion seem at the heart of the argument.

We agree it is extraordinary that specific base changes always arising is strange, though this is not uncommon for viruses, especially given the low, but ubiquitous levels of codon usage bias seen in viruses. Additionally, it is possible that specific nucleotide changes are necessary for regulatory differences. We have discussed this in the manuscript and have also searched for gene conversion and find no evidence of it between the significant SNPs and surrounding SNPs (Figure 5—figure supplement 3). Our argument for increased effective mutation rate coupled to titer also increases the likelihood of this occurring, as we discuss in the manuscript.

In some places the authors appear to assume (or assert) that recombination is frequent, while in others they assume it is completely absent, and nowhere do they explicitly test for it. If it is common, then none of the tree-based analyses can be used. If it is absent, then it is very hard to explain the patterns of diversity or LD in Figure 1—figure supplement 3, and some of the simulations may be inappropriate.

We now address recombination in the second section of the manuscript (after the associations). The reviewer is correct that establishing and being clear about the recombination landscape is crucial for the rest of the manuscript. We have inserted also clarifications in the manuscript in the Discussion. In summary, while recombination events occur anywhere in the genome and frequently, the effective recombination rate is not as extremely high as we may have intimated, as most recombination events are between identical genomes. Recombination between two distinct haplotypes (including the low and high titer haplotypes) would require that the two types infect the same cell and this may be rare even if a host is superinfected. Even when recombination occurs between the two haplotypes, we hypothesize that recombinant genomes may have incompatibilities and so may not be able to leave the host cell. Based on these comments we have also changed our Discussion in the manuscript and have more thoroughly searched for evidence of recombination events between different SNPs. We find no evidence of recombination between populations, meaning each SNP must recurrently evolve in each population, however the final haplotype may have formed by bringing the SNPs onto the same background.

The tree-based analyses seem to be the basis of the major claim that these haplotypes arose independently multiple times, and that the order of the mutations arising was similar. Any tree analysis absolutely requires the absence of recombination or gene conversion, and this needs to be explicitly tested for here (e.g. using GARD, or possibly if diversity is low by inferring the ARG using tsinfer). The failure of LD to decay with distance hints that recombination is absent, but gene conversion over very short distances could still shuffle mutations between haplotypes. However, assuming its branch lengths are in mutations, the shape of the tree in Figure 6A (many short tips below large crowns) shouts 'recombination' to me. Where did this tree come from? It is not ultrametric, so if it was inferred with BEAST this is not standard output.

The reviewer is correct, we have now used GARD to identify recombination events and find evidence of recombination events between significantly associated SNPs and background SNPs. We also see evidence of recombination in our r^2^ analyses between background SNPs but not our significant SNPs. We have also regenerated the phylogeny in BEAST2 using BDsky/skyline (now in Figure 5, appropriate given the recombination findings). We have also generated a phylogeny in BEAST2 considering recombination and get a very similar phylogeny to that seen now in Figure 5. In all cases we find recurrent evolution of each high type SNP in each population, but the combinations of SNPs may either be due to recurrent mutation or recombination. As we find no evidence of recombination between populations, even if the full complement of 11 SNPs has not recurrently evolved on the same background, we still find evidence of recurrent recombination of each mutation in each population.

If the authors do have evidence that recombination is absent, can they confirm that the other analyses (population size history from the SFS; simulations) also assumed zero recombination? Although zero recombination would make me worry even more about the p-values in the GWAS discovery of the SNPs that define the haplotypes – but permutation tests would help deal with that.

To identify recombination events and measure associations of alleles, we have performed the following tests/used the following tools:

– GARD

– R2 in r

– 4 allele test between SNP combinations

We have also factored in recombination in the following analyses and still find support for recurrent evolution of each SNP (though SNPs may have recombined to form the final High Type):

– BEAST2

– TreeTimes

– DeSolve Simulations

– StairwayPlot

We find no evidence of recombination between populations, e.g. we find no recombinant haplotypes which are ½ CH and ½ PR ; however we find recombination within populations. The results shown in Figure 1—figure supplements 2 and 3, suggests recombination is on average quite common. Based on the GARD results and BEAST2 analysis, recombination could have occurred between intermediate types in each population, which may have generated the full High type. However, for this to occur each mutation must recurrently occur in each population on the intermediate type.

I am really confused about their view of recombination, because a couple of times they imply recombination may be common, for example by noting that recombination is required in Nudivirus replication – but then this would invalidate the tree-based analyses? Plus, mechanistic recombination is irrelevant without coinfection infection, since recombination only between identical haplotypes does not have any effect. The observation that no co-infections occur suggests recombination should be very rare – but that's not what the tree looks like. The sixth paragraph of the Discussion suggests that not enough thought has been given to the likelihood, or implications, of recombination.

We think the confusion is the difference between actual recombination (the physical process) and effective recombination (the genetic signature of recombination between two distinct haplotypes). Coinfection is required for effective recombination but not actual recombination. We expect recombination between neutral genotypes to be common if coinfections are common, but do not see recombination between the High and Low types, suggesting either they do not co-infect or recombinants between the two types are inviable. We have included this in the Discussion and reworked our discussion of recombination throughout.

Finally, if there is no recombination, then 'fully derived' haplotypes (those with all 11 SNPs) can only be as old as the youngest of the 11 SNPs, unless the SNPs arose multiple time within populations as well. Is that compatible with the patterns of diversity and the timescale? One check would be to mask those 11, re-infer the tree, and check that the clades are still monophyletic. If these arose by re-current mutations due to strong selection, they could be warping the tree – this might be akin to the problem of recurrent drug or MHC selected mutations in HIV trees, where the known sites are excluded before analysis.If we believe the two haplotypes are real, a much more credible 'story' to me would be two distinct and potentially old haplotypes maintained by selection in the face of a low level of ongoing gene conversion (or recombination). Is there some aspect of the data that this does not fit? This seems just as good a 'story', just as interesting, and just as publishable.

In all cases, the phylogeny has been generated without the 11 focal SNPs but was also generated with the SNPs and is nearly identical in both cases, we have now mentioned this in the manuscript. The idea of the two old haplotypes being maintained in the face of gene conversion is interesting, and we now discuss this in the manuscript. However, the idea of long-term maintenance does not fit with any analyses we have performed, as High types do not cluster together. Additionally, we find the High type is present in the geographically distinct *Drosophila falleni* population and clusters completely separately from the innubila samples.

We do find some evidence of recombination events occurring around the significantly associated SNPs, so it is possible that multiple intermediate strains (with different complements of SNPs) have recurrently evolved in each population and have recombined to generate the full High type, or that the High type has evolved once in each population and is eroded as you described. We address this possibility in both the Results and the Discussion. We now have a dedicated section of our Results regarding detecting recombination and inferring if the High type SNPs were brought together by recombination. We are unable to conclude if the High type was formed exclusively by recombination (likely both sequential mutation and recombination play a role), but find no evidence of recombination between populations, which suggests that even if recombination brings the SNPs together onto the same background, the SNPs must have recurrently evolved in each population.

3) The MK-like analysesAlthough a relatively minor part of the story, the MK analysis is extremely unclear. In part this is because it doesn't appear in the Materials and methods.i) What software was used? The supporting data implies VCFtools can do this (which surprised me) but the method it uses was not explained. Was SNIpre fitted with the original code? The text seem to imply that raw counts were used, which would not be suitable unless Ks was <0.3ii) Although divergence is required, we're not told how (or from what species) it was estimated. If Ks>0.3, then something more than raw counts should certainly be used. If Ks>0.8, I would strongly advise against doing any sort of MK analysis.iii) Figure 4 gives 'difference from background'. What is meant by difference from background? But if the background is a signal of strong constraint, could a positive signal here mean relaxed constraint rather than positive selection. What is the evidence that it's positive selection rather than relaxed constraint?iv) In Figure 4—figure supplement 1 the populations are presented separately, and differences among them are interpreted as differences in positive selection. But surely the divergence number must be massive, larger than the polymorphism number, so almost all of this variation is due to differences in constraint affecting the Pn/Ps ratio. Or is the method one that uses high frequency derived SNPs as evidence of positive selection? If so, how was ancestral state identified? This is probably not possible to do reliably if Ks>0.3. We need to see some raw numbers, and a lot more detail on the methods.

We apologise, the MK-based analyses methods were removed during the writing of the manuscript accidentally, as our results have been reconfigured as 2 manuscripts from one large manuscript, we have added them back. VCFtools was not used, we instead used the original SnIPRE code and SNPeff SNP assignments. We have also explained what we mean by difference from the background, specifically that we found the average statistical measure for nearby genes and compared each focal gene to that, to see how much the focal gene deferred from its surrounding average. We have attempted to answer all your points and questions in this new section of the Materials and methods. Average dS from the outgroups used was 0.133 and 0.279 between DiNV-Kallithea and DiNV-OrNV respectively, which we feel is enough divergence to adequately perform the analysis and is not too much to result in all differences being due to the Pn/Ps ratio. We also used these comparisons to identify the ancestral state, based on the Kallithea and OrNV variant. We have now included our measures of Dn/Ds and Pn/Ps as well as the estimated Selection Effect for our genes of interest as a supplementary figure. In this case we find that envelope proteins and AMPs have an excess of Dn/Ds per site compared to other genes, which is driving their elevated selection effect, while an excess of Pn/Ps and a deficit of Dn/Ds in unknown function DiNV genes is likely driving most other differences. We have chosen to use MK-based tests as opposed to Dn/Ds to identify if the changes are due to positive selection rather than relaxed constraint, as Dn/Ds is weighted by Pn/Ps to identify the proportion of substitutions fixed by selection.

Reviewer #3:[…] 1) Throughout the manuscript, the authors switch back and forth between the present and past tense, which seems awkward from a stylistic point of, I'd suggest to stick to the past tense through-out.

We apologise for this, we have corrected this inconsistency.

2) The fact that the exact same 11 SNPs evolve multiple times independently in mostly the same order is interesting. The authors note that an expected alternative could be different SNPs emerging in the same genes, but they don't discuss the potential mechanism, by which the exact same SNPs seem to be preferred instead.

We have expanded our discussion of the compatibility, both from an evolution perspective and from a functional perspective. First, we hypothesize that the significantly associated SNPs have epistatic effects and thus must occur in a particular order to traverse the fitness landscape form Low type to the High type. In line with this, very few genes show signatures of adaptation in DiNV, which could further limit what beneficial mutations fix. We also discuss the different conditions which could lead to two types of DiNV being found in each population. We also hypothesize some methods these incompatibilities could occur, such as a change in the 19K/PIF-6 protein structure which limits membrane access for viral particles without this variant and could be rescued by a matching change in another protein. Or these two changes together are neutral but in combination prevent membrane access for the other viral type. We also find evidence of recombination events around the significantly associated SNPs, which could easily facilitate the recombination of variants onto the same background, without needing to traverse a lower fitness genotype combination.

3) The authors found little evidence of mixed infection with both the High and Low Types and conclude potential in-compatibility. Please expand on the potential mechanisms of this.

We have expanded our discussion of the compatibility, both from an evolution perspective and from a functional perspective, we have also highlighted similar studies where diverged viruses are unable to coninfect similar systems or are unable to produce viable recombinant particles.

4) Related to the above comment, the authors observe no "hybrids" of High and Low Types again, suggesting "incompatibility". Please discuss the difference between this genetic/genomic versus ecological incompatibility referred to above, as well as the potential link between these two types of incompatibilities.

We have included this in our expanded section in the Discussion. Briefly we discuss how the incompatibility may not be caused by a functional effect, as most SNPs are upstream of genes, but could be due to an effect of the expression of the proteins in combination on viral particle formation, host fitness or even viral ability to transmit between hosts. We do not have zero intermediate types, but we have explained that these could be steps between two fitness peaks as the few intermediates without low fitness.

5) Please rephrase the first sentence of the Discussion (what is the deference between "to better infect" and "optimizing the infection"?).

We have attempted to clarify that we mean that viruses are evolving according to the transmission/virulence trade-off and so are not just evolving to propagate within hosts but to transmit to others, which usually requires the virus not kill the host before transmission can occur.